# The up-regulation of TGF-β1 by miRNA-132-3p/WT1 is involved in inducing leukemia cells to differentiate into macrophages

Zhimin Wang[1,2], Chaozhe Wang [3,4], Danfeng Zhang[5], Xidi Wang[6], Yunhua Wu[4], Ruijing Sun[4], Xiaolin Sun[7], Qing Li[7], Kehong Bi[1]\*, Guosheng Jiang[4,7]\*

1 Department of Hematology, Shandong Provincial Qianfoshan Hospital, Shandong University, Jinan, Shandong, P.R. China, 2 Department of Hematology, Binzhou people's Hospital, Binzhou, Shandong, P.R. China, 3 Department of Blood transfusion, Yantaishan Hospital Affiliated to Binzhou Medical University, Yantai, Shandong, P.R. China, 4 Department of Immunology, College of Basic medicine, Binzhou Medical University, Yantai, Shandong, P.R. China, 5 Department of Laboratory Medicine, Lixia District People's Hospital, Jinan, Shandong, P.R. China, 6 Department of Laboratory Medicine, Zhangqiu District People's Hospital, Jinan, Shandong, P.R. China, 7 Department of Laboratory Medicine, Zibo First Hospital, Zibo, Shandong, P.R. China

\* bikehong11@163.com; jiangguosh@163.com

## Abstract

Although it has been shown that abnormal expression of Wilm's tumor gene 1 (WT1) is associated with the occurrence of leukemia, the specific mechanism via which it induces leukemia cells to differentiate into macrophages remains poorly understood. Based on the prediction that the microRNA miRNA-132-3p is the miRNA that possibly lies upstream of the WT1 gene, we hypothesized that miRNA-132-3p may participate in the polarization process of macrophages through regulating expression of the WT1 gene. The focus of the present study was therefore to investigate the role of the miRNA-132-3p/WT1 signaling axis in the differentiation of THP-1 leukemia cells into macrophages induced by PMA. The results obtained indicated that, compared with the control group, the proliferation of THP-1 cells was clearly inhibited by PMA, and the cell cycle was arrested at G0/G1 phase, associated with an upregulation of CD11b and CD14 expression. Induced by PMA, the expression level of miRNA-132-3p was increased, WT1 expression was decreased, and the expression level of TGF-β1 was increased. Following transfection with miRNA-132-3p mimics, however, the expression of WT1 in the THP-1 cells was downregulated, with upregulation of the CD11b and CD14 antigens, whereas this downregulation of WT1 mediated by miRNA-132-3p mimics could be reversed by co-transfection with WT1 vector, which was accompanied by downregulation of the CD11b and CD14 antigens. The luciferase activity of the co-transfected miRNA-132-3p mimic+WT1-wild-type (WT) group was found to be statistically significantly lower compared with that of the co-transfected miRNA-132-3p mimic+WT1-mutated (MUT) group. Furthermore, chromatin immunoprecipitation experiments showed that WT1 was able to directly target the promoter of the downstream target gene TGF-β1, which led to

**Data availability statement:** All relevant data are within the manuscript and its Supporting Information files.

**Funding:** 1. Natural Science Foundation of Shandong Project Number: ZR2021MH080 Project Undertaker: Jiang Guosheng Role of the funder: ChIP assay and detection of gene expression 2. Cultivation Fund of the first affiliated hospital of Shandong First Medical University Project Number: QIPY2020NSFC0819 Project Undertaker: Bi Kehong Role of the funder: Establishment of Cell Differentiation Model, Western blotting assay, detection of gene expression and Luciferase Assay 3. The Foundation for Jinan's Clinical Science and Technology Innovation Project Number: 202134001 Project Undertaker: Zhang Danfeng Role of the funder: transfection assay, Western blotting assay and detection of gene expression 4. Shandong Province Medical and Health Technology Development Plan Project Project Number: 202003041248 Project Undertaker: Wang Zhimin Role of the funder: Flow cytometry assay, Western blotting assay and detection of gene expression 5. Scientific research project of Binzhou People Hospital Project Number: XJ2023000305 Project Undertaker: Wang Zhimin Role of the funder: Flow cytometry assay, Western blotting assay 6. Shandong Province Medical and Health Science and Technology Project Project Number: 202411000179 Project Undertaker: Wang Xidi Role of the funder: Detection of gene expression and protein expression Project Undertakers including Wang Zhimin, Zhang Danfeng, Wang Xidi, Bi Kehong and Jiang Guosheng. Wang Chaozhe, Wu Yunhua, Sun Ruijing, Sun Xiaolin and Li Qing are participants in the above-mentioned projects.

**Competing interests:** The authors have declared that no competing interests exist.

the negative modulation of TGF-β1 expression, whereas downregulation of WT1 led to an upregulation of the expression of TGF-β1, which thereby promoted the differentiation of THP-1 cells into macrophages. Taken together, the present study has provided evidence, to the best of the authors' knowledge for the first time, that the miRNA-132-3p/WT1/TGF-β1 axis is able to regulate the committed differentiation of leukemia cells into macrophages.

## Introduction

Acute myeloid leukemia (AML) is a common malignant hematological tumor originating from promoter cells [1–3]. In clinical practice, cytotoxic chemotherapy regimens remain the predominant method for treatment of patients with AML [4]. However, resistance to these chemical drugs often develops; moreover, damage frequently occurs to the normal hematopoietic and immune cells, which motivates researchers to find safer and more effective treatment methods. It has been shown that the occurrence of leukemia is to a large extent caused by the differentiation block of pluripotent hematopoietic stem cells at a certain stage during the maturation process, and patients with leukemia could be treated with differentiation inducers to induce the leukemia cells to differentiate into more mature cells [5,6]. For example, all-trans retinoic acid (ATRA) has been used to treat patients with acute promyelocytic leukemia (APL) [7], and different leukemia cell lines have been shown to differentiate into mature monocytes, erythrocytes or granulocytes, depending upon their induction by different inducers [8–12]. However, in spite of the success of ATRA in inducing APL cells to differentiate into granulocytes, to date, no breakthroughs have been achieved in terms of pathways for the directed differentiation of monocytes or macrophages.

As a result, the identification of novel differentiation intervention targets or differentiation-inducing drugs is of crucial importance, and studying the underlying molecular mechanism of abnormal cell differentiation will provide the basis for discovering novel targeted differentiation-inducing drugs. After performing a literature review, we found that Wilm's tumor gene 1 (WT1) is closely associated with the occurrence and development of solid tumors or hematological tumors, suggesting that it may be involved in the committed differentiation process of leukemia cells. WT1 is located on chromosome 11p13, and encodes a DNA-binding protein [13]. The carboxyl-terminus of the protein encoded by this gene contains four zinc finger domains, which acts a DNA-binding functional region. It is able to identify and bind to the GC-enriched fragment (CCCCCGC) of the target gene in its promoter region, whereas the amino-terminus is rich in glutamine/proline residues, mainly serving a role in transcriptional regulation [14,15]. It has been reported that WT1 is an effective transcription factor that exerts multiple roles in the proliferation, differentiation, development and apoptosis of normal and tumor cells [16,17]. To date, studies have shown that WT1, as a tumor gene, is usually upregulated in solid tumors or hematological malignancies [18], and WT1 protein has a crucial role as a regulator of

transcription, participating in the proliferation, differentiation and apoptosis of leukemia cells [19,20], as well as having an association with the diagnosis, minimal residual disease (MRD) and prognosis of patients with leukemia [21–22]. Furthermore, the high expression of WT1 provides useful opportunities for the development of vaccines, targeted antibodies or chimeric antigen receptor immunotherapy [23,24]. However, the exact role of WT1 in the development of leukemia remains poorly understood; moreover, the complexity associated with this function of WT1 will require a great research effort in the future [25]. Even though some research groups have found that the expression of WT1 is downregulated in leukemia cells induced by differentiation inducers [26–28], to the best of the authors' knowledge, few studies have been published on either the committed differentiation of WT1 in leukemia, or on the committed differentiation of leukemia cells into macrophages.

On the other hand, numerous studies have shown that oncogenes are often regulated by upstream noncoding RNA or transcriptional regulatory proteins; for example, microRNAs (miRNAs) usually negatively modulate the expression of their target genes, or otherwise regulate important biological processes, such as cell differentiation and apoptosis [29–31]. We hypothesized that a certain miRNA may be involved in regulating the expression of WT1. Among these miRNAs, miR-132 is located on chromosome 17, and consists of two homologous miRNAs, namely hsa-miR132-5p and hsa-miRNA-132-3p. Although it has been demonstrated that miR-132 may be a valuable biomarker candidate for the prognosis of various solid tumors [32–41], to date, few studies have been published on its role in leukemia. Preliminary experiments and database analyses have indicated that miRNA-132-3p is associated with the occurrence of leukemia, and it is also a possible upstream miRNA that targets the 3'-untranslated region (3'-UTR) of WT1. However, further in-depth studies are required to delineate how it may regulate WT1, and modulate the differentiation of leukemia cells. In addition, transforming growth factor-β (TGF-β) has also been shown to regulate various cellular activities, including apoptosis, differentiation and angiogenesis, and TGF-β is known to participate in the occurrence of tumors [42,43] and in leukemia [44]. Furthermore, TGF-β1 can interact with the receptor of TGFβ1 to participate in cell proliferation, differentiation and apoptosis, or to promote the differentiation of leukemia cells [45]. Interestingly, through database analysis and differential gene chip screening, we found that WT1 may be a target gene for hsa-miRNA-132-3p, and that WT1 is associated with downstream TGF-β1 in the pathway.

However, the role of the miRNA-132-3p/WT1/TGF-β1 axis in inducing leukemia cells to differentiate into macrophages has yet to be fully elucidated. Therefore, the aim of the present study was to investigate whether there is a targeted relationship, and any regulatory effects, between miRNA-132-3p and WT1, and subsequently to determine whether WT1 is target-associated with TGF-β1, and whether it has a regulatory effect on both miRNA-132-3p and TGF-β1 during the committed differentiation of leukemia cells induced by PMA, in order to ascertain whether the miRNA-132-3p/WT1/TGF-β1 signaling axis is involved in the process of differentiation of leukemia cells into macrophages.

## Materials and methods

### Materials

Three human leukemia cell lines (including the acute monocytic leukemia cell lines U937 and THP-1 and the chronic myelocytic leukemia cell line K562; Shanghai Cell Bank, Chinese Academy of Sciences) were employed for the following experiments. PMA (Sigma-Aldrich; cat. no. P1585-1MG) was used as the differentiation inducer. FBS, the BCA protein assay kit and SDS-PAGE gel preparation kit were obtained from Shanghai Biyuntian Biotechnology Co., Ltd. HyClone® RPMI-640 Complete™ medium was purchased from Cytiva. Wright-Giemsa staining solution, the secondary antibody, RIPA protein lysis solution and 5X protein loading buffer were purchased from Beijing Solarbio Science & Technology Co., Ltd. The standard protein marker and the Lipofectamine® 2000 transfection kit were purchased from Thermo Fisher Scientific, Inc. The ECL luminescence kit was purchased from Shandong Sparkjade Scientific Instruments Co., Ltd. Regarding the antibodies, WT1 antibody (cat no. 83535S) was purchased from CST Biological Reagents Co., Ltd., β-actin (cat. no. 20536–1-AP) primary antibody was from Proteintech Group, Inc., TGF-β1 antibody (cat. no. YT4632)

was purchased from Immunoway Biotechnology, Inc., and HRP-labeled rabbit secondary antibody (cat. no. ZB-2301) was provided by OriGene Technologies, Inc. Sangon Biotech Co., Ltd. designed and synthesized the primers. The Reverse transcription (RT) kit, PrimeScript™ RT kit with gDNA eraser (for perfect real-time experiments), chimeric fluorescence detection kit and TB Green Premix Ex Taq™ (Tli RNaseH Plus) kit were obtained from Takara Biomedical Technology (Beijing) Co., Ltd. Cell cycle analysis kit was purchased from Jiangsu KGI Biotechnology Co., Ltd. The fluorescent binding antibodies PE-CD11b (cat. no. 301306) and FITC-CD14 (cat. no. 301804) were purchased from BioLegend, Inc. The TRIzol® reagents, which were manufactured by Thermo Fisher Scientific, Inc., were purchased from Tiangen Biotech Co., Ltd. Finally, trichloromethane was purchased from Sinopharm Chemical Reagent Co., Ltd. Cell counting kit-8 (CCK8) was purchased from DOJINDO (Cat.no.CK04). Inhibition of proliferation of leukemia cells induced by PMA. THP-1, U937 and K562 cells were grown in culture flasks containing 10% FBS in RPMI-1640 Complete™ medium, cultured at 37°C in an atmosphere containing 5% CO2. When the three types of cells had reached the exponential growth phase, they were collected and added to the 96-well plates, and the PMA solution was also added, so that the final cell concentration in each well was $1\times10^5$/ml, and the corresponding concentration of PMA solution was 100 ng/ml (based on the $IC_{50}$ concentration, where the $IC_{50}$ value represents half-maximal inhibitory concentration). An equal volume of 100 µl RPMI-1640 medium was used as the negative control group. Three duplicate wells were set up for each group, and the three types of cells were subsequently cultured at 37°C in an atmosphere of 5% $CO_2$ and with saturated humidity. After 48 h, 10 µl CCK8 solution was added to each well and incubated at 37°C for a further 2 h, and then the OD values of the 96-well plate at 450 nm were measured.

## Wright-Giemsa staining to observe morphology of the THP-1 cells

After the THP-1 cells were induced with 100 ng/ml PMA *in vitro* for 48 h, the morphological changes of the cells were observed directly under an inverted optical microscope, and Wright-Giemsa staining was employed to stain the cells both before and after their exposure to PMA. The main steps of the procedure were as follows: First, $2\times10^5$ THP-1 cells of the above-mentioned control and experimental groups were collected after exposing the cells to PMA for 48 h. The cells were subsequently washed twice with cold PBS and resuspended in 100 µl PBS, preparing a cell suspension. The cell suspension was centrifuged at 205 x g for 3 min at 4°C, after which the centrifuged smear was dried and stained with Wright-Giemsa staining solution at room temperature for 5 min. Finally, changes in cell morphology were observed under an optical microscope, and the resulting images were captured with full field photography.

## Flow cytometry to detect the expression of the CD11b and CD14 antigens

The THP-1 cell suspension was collected for the control group and different experimental groups, placed into test tubes, and the cells were then resuspended in PBS buffer, before washing twice with PBS and centrifuging the cell suspensions at 205 x g for 3 min at 4°C. The cell concentration was adjusted to $1\times10^6$/ml, and 100 µl PBS was added into each test tube. Subsequently, 5 µl PE-labeled mouse anti-human CD11b fluorescent antibody (cat. no. 301306; 1:1,000 dilution; BioLegend, Inc.) or FITC-labeled mouse anti human CD14 fluorescent antibody (cat. no. 301804; 1:1,000; BioLegend, Inc.) was added into each tube, respectively. The cells were incubated in the dark at 4°C for a further 30 min, and then resuspended with 200 µl PBS and fixed in 1% paraformaldehyde at 4°C for 20 min. A FACSVerse flow cytometer (BD Biosciences) was used to measure the expression levels of the CD11b and CD14 antigens. Isotypic rat IgG (BioLegend, Inc.) was used to examine for nonspecific binding. All experiments were repeated three times, and the expression of CD11b and CD14 was evaluated as the mean ratio.

## Western blot assay to detect expression of WT1 and TGFβ1

Chiefly, $3\times10^6$ THP-1 cells were first collected from the control group and different experimental groups, an appropriate amount of pre-cooled PBS buffer was added, and the cell mixture was centrifuged at 462 × g for 5 min at room

temperature, before washing the cells twice with PBS buffer. After discarding the supernatant, the cells were lysed using RIPA buffer (Beijing Sun Biotechnology Co., Ltd.) to extract total cell proteins, and the protein concentration was then determined using the BCA method. Subsequently, 5X protein loading buffer was added, and the proteins were denatured by boiling at 95°C for 10 min. Aliquots (20 µg) of protein were then loaded on to SDS-PAGE gels to perform SDS-PAGE (10% gels), and the separated proteins were then transferred onto a PVDF membrane. Subsequently, the membrane was sealed with 5% skimmed milk at room temperature for 90 min, prior to incubation with the corresponding primary antibody. WT1 (1:1,000), TGF-β1 (1:10,000) and GAPDH (1:2,000) antibodies were added, and the primary antibodies were incubated with the membrane overnight at 4°C. The membranes were washed with 1X TBST for 30 min. Subsequently, after the first antibody reactions were completed, the membranes were exposed to goat anti-rabbit IgG secondary antibody (1:20,0000 dilution) for an incubation at room temperature for 90 min. Finally, the membranes were washed with 1X TBST for 30 min. The proteins were detected using an ECL kit (Sparkjade ECL Super; cat no. ED0015-B; Shandong Sparkjade Scientific Instruments Co, Ltd,), and ImageJ, 1.51j8 software was used to determine the densities of the protein bands. All experiments were repeated three times, and the results were expressed as the average relative expression densities.

## Databases used to predict the association between miRNA-132-3p and WT1

Three databases (TargetScan (http://www.targetscan.org), PITA (http://genie.Weizmann.ac.il/pubs/mir07/mir07_data.html) and microRNAorg (http://www.microrna.org/), were selected to accurately predict the possible upstream miRNAs of WT1. Based on the potential miRNAs predicted by each database, a cross-analysis of the three databases was conducted. Through an analysis of the intersection of the three databases, common target miRNAs were obtained, and the most likely potential miRNAs to target WT1 were selected on the basis of the size of the P-value. In addition, differential miRNAs between the PMA experimental group and negative control group were detected and analyzed using the GeneChip miRNA 4.0 detection chip (Affimatrix Co., Ltd.) The types of miRNAs that were upregulated after induction of differentiation were screened based on their P-values. In addition, the abovementioned databases were also used to predict miRNAs that were associated with macrophage differentiation. The upregulated miRNAs identified by crosstalk from the mentioned three different analyses were selected to represent the potential miRNAs for targeting WT1.

## RT-quantitative (RT-q) PCR assay to detect the expression of *miRNA-132-3p* and *WT1*

The THP-1 cells were observed under an inverted microscope until they had reached the exponential phase of growth. Subsequently, 1x10^6 cells were collected, and the total RNA was extracted using TRIzol°, following the instructions provided by the kit (Thermo Fisher Scientific, Inc.). After the total RNA concentration had been measured using a Ultra-micro Nucleic Acid Protein Analyzer, the cDNA was synthesized using an RT kit [specifically, according to the instructions of the PrimeScript° RT kit and the gDNA Eraser (Perfect Real Time) RT kit]. The main steps of the RT reaction were carried out; for example, the sample was incubated in the Eppendorf PCR system at 42°C for 15 min, after which it was incubated at 85°C for 5 secmin, and then incubated at 4°C for a further 2 min. The cDNA obtained by RT was then used as the next PCR amplification template. The sequence of the miRNA-132-3p sense strand was 5'-CGATACCGTTCTAACAG TCTACAGC-3', whereas that of the antisense strand was 5'-TATGGTTTTC ACGACTGTGTGAT-3'; the sequence of the U6 sense strand was 5'-CTCGCTTC GGCAGCACA-3', and that of the antisense strand was 5'-AACGCTTCACGAATT TGCGT-3', and these primers were synthesized by Shanghai GenePharma Co, Ltd. These two primers were designed according to the stem-loop method. The downstream and RT primers of the stem-loop method were artificially added (Shanghai Sangon Biotech, Co., Ltd.). The WT1 sense strand was 5'-CAGGCTGCAATAAGAGATATTTTAAGCT-3', and the antisense strand was 5'-GAAGTCACACTGGTATGGTTTCTCA-3'; the GAPDH sense strand was 5'-CAACTTTGG TATCGTGGAAGG-3', and the antisense strand was 5'-GCCATCACGCCACAGTTTC-3', and these were synthesized by Biosune Biotechnology (Shanghai) Co., Ltd. The real-time fluorescence quantitative amplification reactions were

performed mainly following the instructions of the TB Green® Premium Ex Taq (Tli RNaseH Plus) kit. The thermocyling conditions of the RT-qPCR process mainly comprised the following key steps: 10 sec at 95°C; 5 sec at 60°C, and 10 sec at 72°C for a total of 40 cycles, with a final step of 60°C for 34 sec. The results were quantified with the $2^{-\Delta\Delta Cq}$. These experiments were repeated three times, and the results were expressed as the mean ± standard deviation.

### Dual-luciferase reporter gene assay for miRNA-132-3p targeting of WT1

The regional sequence of WT1 gene for miRNA-132-3p binding was predicted using the 'TargetScan' bioinformatics website (http://www.targetscan.org). It was entrusted to Biosune Biotechnology (Shanghai) Co., Ltd. to construct the wild-type plasmid vector pmirGLO-WT1-wt, and the mutant plasmid vector pmirGLO-WT1-mut (www.Biosune.com). 293T cells (as a density of $7x10^4$ cells/well) were added to the wells of a 24-well culture plate, and subsequently incubated for further 24 h at 37°C in an atmosphere of 5% $CO_2$ and with saturated humidity. Following the instructions provided in the cell transfection kit (jetPRIME; Polyplus Co., Ltd), the WT1 wild-type plasmid, the WT1 mutant plasmid and miRNA-132-3p mimics were co-transfected into the 293T cells. After further cultivation for 48 h, the luciferase activity was measured using a dual-luciferase reporter assay system (EnVision™; PerkinElmer Co., Ltd.), according to the manufacturer's instructions. Data were normalized for transfection efficiency by dividing the firefly luciferase activity with that of Renilla luciferase. During the 48 hour process of PMA induced differentiation of THP1 cells in vitro, 7 different time points were randomly selected, and qRT PCR was used to measure the changes in gene expression levels of miRNA 1323p and WT1, and Pearson correlation analysis was performed to assess the correlation between miR-132-3p and WT1.

### Cell transfection assay

Cell transfection reactions were performed following the instructions provided by the kit. For the short interfering RNA (siRNA) interference experiment, FAM fluorescence-labeled si-NC was used as the negative control in parallel to determine the transfection efficiency. According to the instructions of the transfection kit, the siRNA was mixed with the transfection reagent to form a transfection complex. At the same time, the corresponding negative control si-WT1 scramble NC(or scrambled siRNA) was also transfected. After that, their transfection complexes were added the inoculated cells in a 6-well plate, and the cells were then cultured for 48 h at 37°C for the subsequent associated detection experiments. The sequence of the si-WT1 sense strand was 5'-CUACAGCAGUGA CAAUUUAUT-3', whereas that of the si-WT1 antisense strand was 5'-UAAAUUGU CACUGCUGUAGTT-3'. The sequence of the si-WT1 scramble NC sense strand was 5′-AUU GUAAUCCACAAGUGCATT-3′, and that of its antisense strand was 5′-UGCACUUGUGGAUUACAAUTT′. The siRNAs were synthesized by Shanghai GenePharma Co., Ltd.

Regarding the siRNA interference experiments, four experimental groups were set up to perform the siRNA interference experiments, as follows: the 0 µg/l control group, the 100 µg/l PMA experimental group, the 100 µg/l PMA + si-NC transfection experimental group, and the 100 µg/l PMA + si-WT1 transfection experimental group. The si-WT1 transfer methods were followed as per the instructions provided with the transfection kit. In addition, the role of the miRNA-132-3p mimic in modulating WT1 expression was also investigated. The sequences of the primers for the miRNA-132-3p mimic were as follows: Sense, 5'-UAACAGUCUACAGCCAUGGUCG-3'; and antisense, 5'-CGACCAUG GCUGUAGACUGUUA-3'; and the sequence of the mimic NC sense strand was 5'-UUCUCCGAACGUGUCACGUTT-3', whereas that of the antisense strand was 5'-ACGUGACACGUUCGGAGAATT'3'. These primers were also synthesized by Shanghai GenePharma Co, Ltd., and the transfection experiments were performed using Lipofectamine® 2000, following the instructions provided by the kit. miRNA-132-3p mimic was transfected into the THP-1 cells, and RT-qPCR and western blotting were performed to detect the expression levels of miRNA-132-3p and WT1 protein, respectively, as described above. At the same time, the corresponding expression levels of CD11b and CD14 of THP-1 cells were detected by flow cytometric assay.

 

## Chromatin immunoprecipitation (ChIP) assay

ChIP assay was performed to detect the targeted binding and regulation of TGF-β1 by WT1. After induction by PMA for 2h, a total of $2x10^7$ THP-1 cells were collected, and subsequently formaldehyde at a final concentration of 1% was added into the tubes, and the mixture was incubated further for approximately 10min. After the preparation of cross-linked chromatin had been completed, ultrasound treatment was used to achieve an average size of DNA fragments of 200–1,000bp, as determined by ethidium bromide gel electrophoresis. The purified chromatin was immunoprecipitated using 3 μg WT1-specific antibodies or an irrelevant antibody (IgG). The immunoprecipitates were kept overnight at a temperature of 4-°C-. Finally, the immunoprecipitated chromatin was used as the template for amplification by PCR. Annealing reactions were performed at 59.2°C, and the PCR products are separated by routine agarose gel electrophoresis. The primers for TGF-β1 were as follows: Sense, 5'-ATTAAGCCTTCTCCGCCTGGTCCT-3', and antisense, 5'-CGGCAACGGAAAA GTCTCAAAAGT-3' [Biosune Biotechnology (Shanghai) Co, Ltd.].

## Rescue experiment of miRNA-132-3p by si-TGF-β1 upon inducing differentiation of the THP-1 cells

Regarding the si-TGF-β1 interference experiments for reversing the effects of miRNA-132-3p in inducing differentiation of the THP-1 cells, three experimental groups were set up; namely, the miRNA-132-3p mimics-NC, miRNA-132-3p mimics and miRNA-132-3p mimics+siTGF-β1 groups. Following the instructions provided by the manufacturer of the kit for the Lipofectamine 2000™ transfection assay, the miRNA-132-3p mimics, siTGF-β1 and their si-NCs were transfected into the THP-1 cells. At 20h post-transfection, the cells were collected, and the expression of TGF-β1 protein was detected by western blotting assay, whereas the expression levels of CD11b and CD14 were detected by FACS at the same time. We have designed four pairs of si-TGFβ1, and the interference sequence was validated for interference efficiency through qRT-PCR and Western blotting experiments, and the pair of sequences with the most obvious interference efficiency was selected as the candidate siRNA in the present manuscript.The sequence of this candidate siTGF-β1 sense strand was 5′-CACUGCAAGUGGACAUCAATT-3′, whereas that of the antisense strand was 5′-UUGAUGUCCACUUG CAGUGTT-3′. The si-TGF-β1 scramble NC sense strand was 5′-ACUUCAACGCCAGUGAAAGTT-3′, whereas that of the antisense strand was 5'-CUUUCACUGGCGUUGAAGUTT-3' (Shanghai GenePharma Co., Ltd.). The sequences of the miRNA-132-3p mimics-NC and miRNA-132-3p mimics primers were the same as mentioned above.

## Statistical analysis

The data were processed using GraphPad Prism 8 software, and the quantitative data were analyzed using Shapiro Wilk (S-W) normal distribution. The differences between two groups were compared using independent sample t-tests. ANOVA analysis was used for making comparisions with multiple groups. One-way ANOVA was performed using the Bonferroni test. Pearson correlation analysis was performed to assess the correlation between miR-132-3p and WT1. Double-tailed tests were also employed for analysis of all the data. Each experiment was repeated three times, and the results were expressed as the mean±standard deviation. $P < 0.05$ was considered to indicate a statistically significant difference.

## Results

### Cell growth density and proliferation of leukemia cells induced by PMA

Induced by PMA at a concentration of 100ng/ml, the K562, U937 and THP-1 cells were all found to be significantly inhibited by PMA as compared with that of their respective control groups ($0.67 \pm 0.01$ vs. $0.43 \pm 0.02$, t=22.88, $P < 0.001$; $0.67 \pm 0.01$ vs. $0.34 \pm 0.02$, t=28.81, $P < 0.0001$; and $0.67 \pm 0.01$ vs. $0.32 \pm 0.02$, t=17.22, $P < 0.0001$) (Fig 1A). Based on the result that the most significant inhibitory effect on cell proliferation was observed in THP-1 cell line, which was selected to construct a model of committed differentiation from leukemia cells into macrophages. After exposure to PMA for 48h, the cell density of the THP-1 cells was significantly reduced as observed under a microscope (Fig 1B),

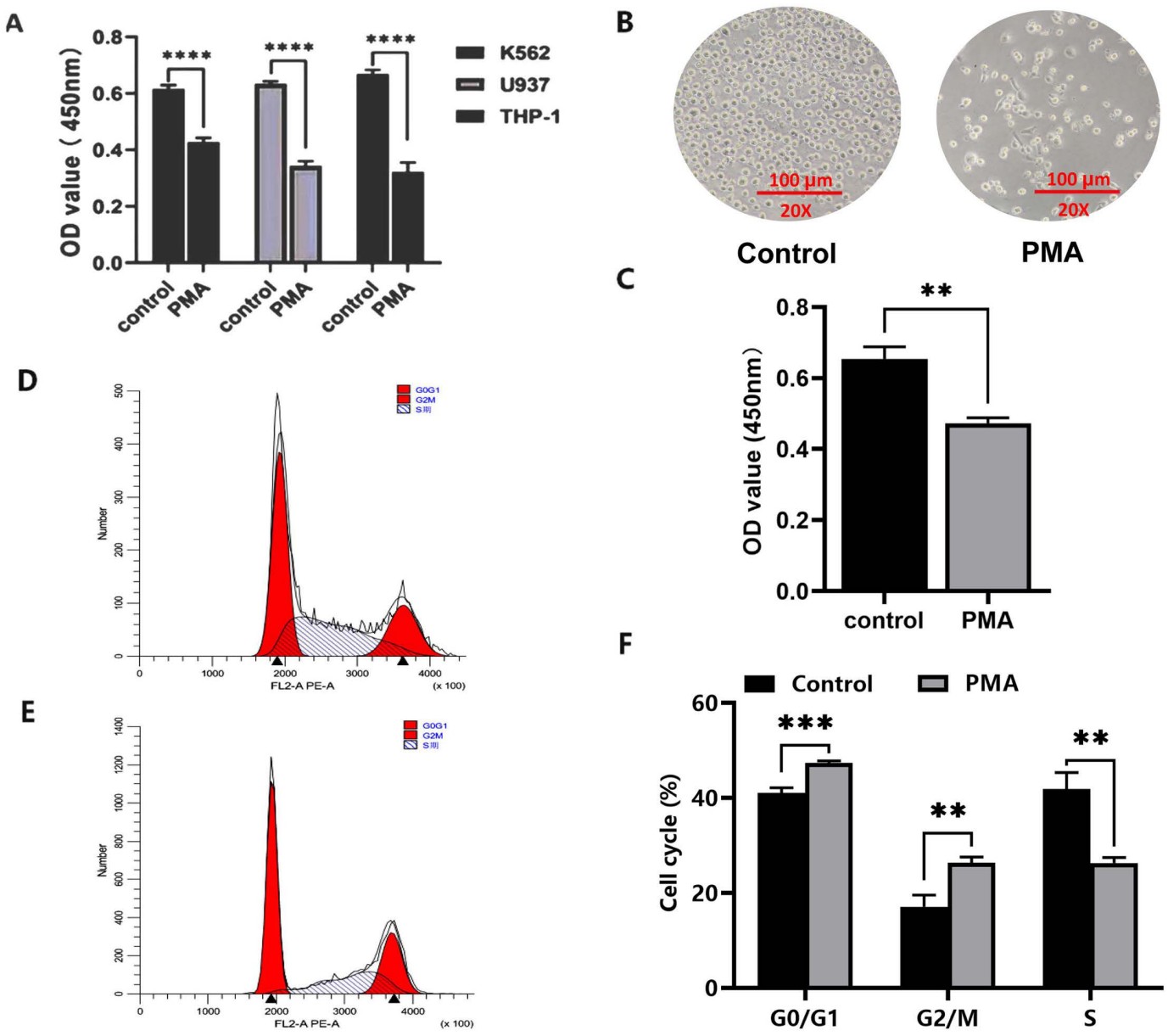

**Fig 1. The proliferation and cell cycle analysis of leukemia cells after exposure to PMA.** K562, U937 and THP-1 leukemia cells were treated with PMA 100 ng/mL for 48 hours, the cell density were directly observed under a microscope, and proliferation of these leukemia cells were detected by CCK8 cell counting assay. Cell cycle was detected by Flow cytometry assay. A: Proliferation of K562, U937 and THP-1 leukemic cells before and after exposure to PMA; B. Cell density of THP-1 cells under microscopy; C: CCK8 assay to detect the proliferation of leukemia THP-1 cells; D. Cell cycle changes of control group THP-1 cells by flow cytometry assay; E: Cell cycle changes of PMA group cells by flow cytometry assay; F: The average percentage of cell cycle distribution. Each experiment was repeated three times. (* $P<0.05$, ** $P<0.01$, **** $P<0.0001$).

and the proliferation of the THP-1 cells was significantly inhibited by PMA ($0.65\pm0.03$ vs. $0.47\pm0.01$, $t=8.522$, $P<0.01$) (Fig 1C). The flow cytometry analysis showed that the G0/G1 phase of THP-1 cells increased from 45.44 to 52.52% ($45.44\pm0.77\%$ vs. $52.52\pm0.68\%$, $t=11.860$, $P=0.0003$), whereas that of the S phase decreased from 31.55 to 15.72% ($31.55\pm1.35\%$ vs. $15.72\pm1.27\%$, $t=14.77$, $P=0.0001$), and the G2/M phase was increased from 23.01% to 31.76%

(23.01 ± 1.52% vs. 31.76 ± 1.63%. t = 6.808, *P* = 0.0024). Taken together, these experiments showed that the proliferation of THP-1 cells could be inhibited by PMA, and that PMA could induce cell cycle arrest at G0/G1 phase and G2/M phase (Fig 1D-1F).

## Expression of the CD11b and CD14 differentiation antigens of the leukemia THP-1 cells

On the basis of the cell proliferation experiments which confirmed that PMA could significantly inhibit the proliferation of THP-1 cells, further experiments were performed to detect the committed differentiation level of this type of cells into macrophages. Usually, in the committed differentiation experiment of leukemia cells into monocytes and macrophages, the quantitative detection of CD11b and CD14 differentiation antigens is mainly used. However, in order to more intuitively demonstrate the successful model of leukemia cells to differentiate into monocytes and macrophages, Wright-Giemsa staining was used to observe differentiation of THP1 cell induced by PMA. As compared with the control group (Fig 2A), after exposure to PMA *in vitro* for 48 h, the THP-1 cells were observed to have increased in size, and the ratio of cytoplasm to nucleus was increased, moreover, certain cells exhibited two or even multiple nuclei, showing a trend of differentiation towards macrophages (Fig 2B), and the population of more matured differentiated THP-1 cells had clearly increased (Fig 2C) (*P* < 0.001). The expression levels of the CD11b and CD14 antigens, which acted as macrophage-specific markers on the cell surface of the THP-1 cells, were also detected by flow cytometry. The expression of CD11b was found to be upregulated to a greater extent compared with that in the control group (35.76 ± 1.76% vs. 2.59 ± 0.03%, t = 32.70, *P* < 0.0001) (Fig 2D and 2E), and the expression of CD14 was also higher compared with that in the control group (26.27 ± 0.84% vs. 3.60 ± 0.52%, t = 68.73, *P* < 0.0001) (Fig 2F and 2G). Therefore, both examining the morphological changes into more mature cells, and detecting the upregulation of CD11b and CD14 expression served to demonstrate that THP-1 cells could be induced by PMA towards differentiation into macrophages.

## Role of WT1 in PMA-induced committed differentiation of THP-1 cells

In our pre-experimental studies, the KEGG and BIOCARTA databases were used to analyze and screen genes associated with PMA-induced cells, and a Pubmed/MEDLINE literature research was performed to identify the oncogenes that were associated with proliferation of the leukemia cells. After crosstalk analysis, it was found that WT1 was involved in the proliferation of leukemia cells, and therefore our attention turned to detecting the role of WTI in regulating the differentiation of THP-1 cells into macrophages. Firstly, WT1 expression in the THP-1 cells was downregulated following exposure to PMA 1.00 ± 0.11 vs. 0.62 ± 0.05, t = 5.392, *P* = 0.006) (Fig 3A and 3B). To further confirm that PMA works really by regulating the expression of WT1, WT1 transfection experiment was conducted,and the results showed that empty vector transfection did not affect the effect of PMA on modulating WT1 expression (t = 0.6224, *P* = 0.5674) (Fig 3C, 3D) However, when transfected with WT1 vector after exposure to PMA, the downregulation of WT1 expression was significantly reversed by WT1 vector transfection (t = 7.164, *P* = 0.0020) (Fig 3E and 3F). When compared with the control group, the mean rate of decrease in the expression level of WT1 mediated by PMA was 60.20 ± 6.93%, whereas the mean rate of increase in WT1 expression for the WT1 vector + PMA group was 159.60 ± 7.60% (Fig 3G). On the basis of confirming that WT1 vector can reverse the downregulation of WT1 expression by PMA, we further validated whether this transfection affects the differentiation level of THP1 cells. It was also found that the WT1 vector transfection could reverse the the downregulation of WT1 expression by PMA (0.59 ± 0.09 vs. 0.92 ± 0.14, t = 3.416, *P* = 0.0259) (Fig 3H and 3I). Reversal of PMA-induced downregulation of WT1 by transfection with WT1 vector (Fig 3I), the percentages of CD11b- or CD14-positive cells in the WT1 vector + PMA group were downregulated compared with those in the PMA-induced group alone, such as CD11b antigen (13.85 ± 0.43% vs. 22.02 ± 0.23%, t = 29.27, *P* < 0.0001) (Fig 3J, 3K) and CD14 antigen(19.06 ± 2.61% vs. 32.13 ± 2.69%, t = 6.028, *P* = 0.0038) (Fig 3L,M). Taken together, these results suggested that overexpression of WT1 is able to counteract the differentiation-induced effects on the THP-1 cells mediated by PMA.

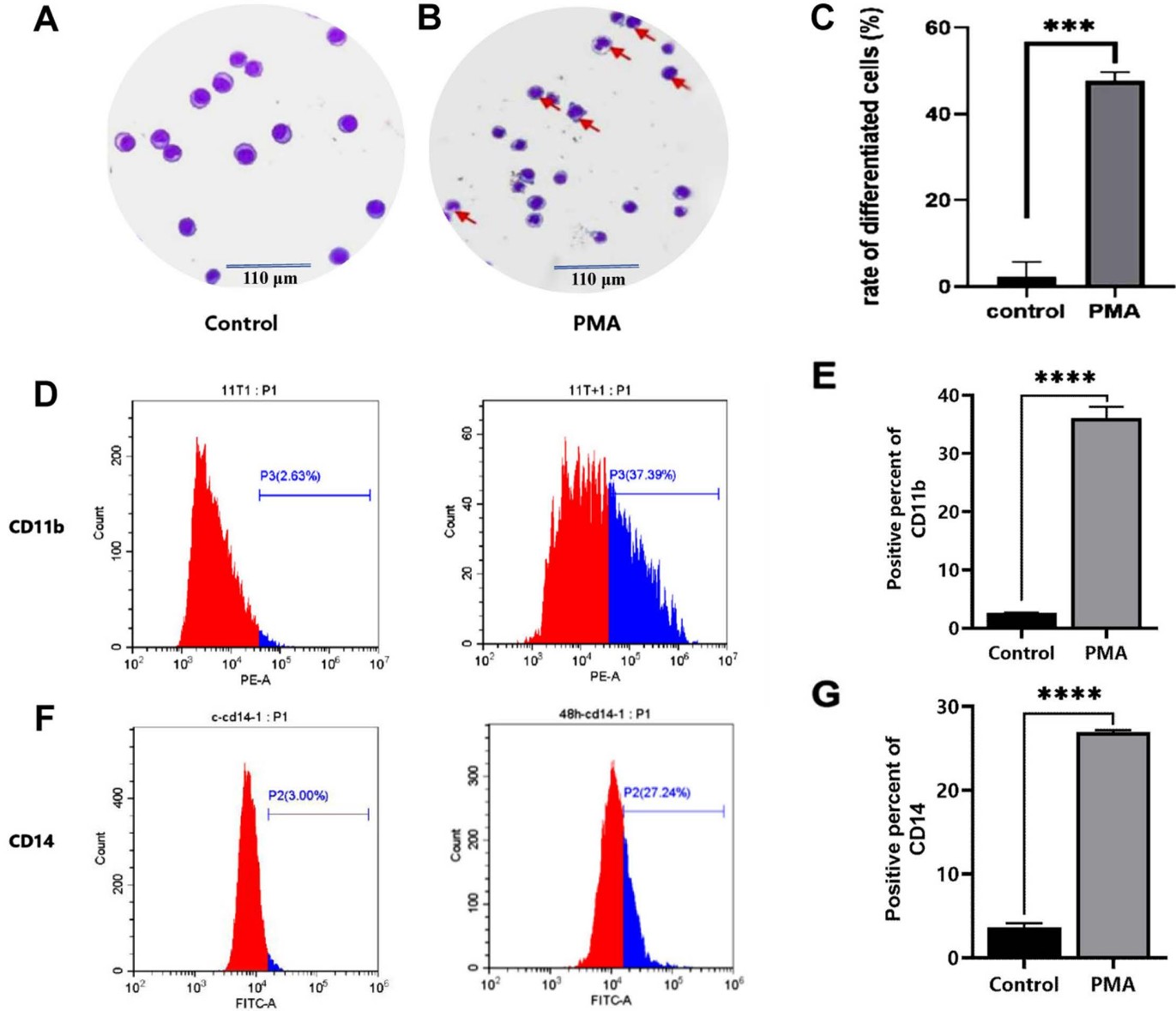

**Fig 2. Effects in differentiation of THP-1 leukemia cells after PMA treatment.** Induced by PMA for 48 hours, the THP-1 cells were collected and stained with Wright-Giemsa staining, and expression of CD11b and CD14 antigens on the cell surface was detected by flow cytometry. A. Wright-Giemsa staining of THP-1 cells control group; B. Wright-Giemsa staining of THP-1 cells induced by PMA; C. The mean rate of differentiated THP-1 cells; D: Expression changes of CD11b expresson percent in THP-1 cells in control group and PMA-induced group by flow cytometry assay; E. Mean expression of CD11b in control group and PMA-induced group THP-1 cells; F. Expression changes of CD14 expresson percent in control group and PMA-induced group by flow cytometry assay; G: Mean CD14 expression percent in control and PMA-induced group. Each experiment was repeated three times. (**** $P < 0.0001$).

## Cross analysis of potential miRNAs targeting WT1, and their associations during committed differentiation towards macrophages

The putative targets of miRNA-132-3p were predicted by analysis of the TargetScan (http://www.targetscan.org/) and miR-TarBase (http://mirtarbase.mbc.nctu.edu.tw/php/index.php) databases, also with the aim of predicting potential upstream

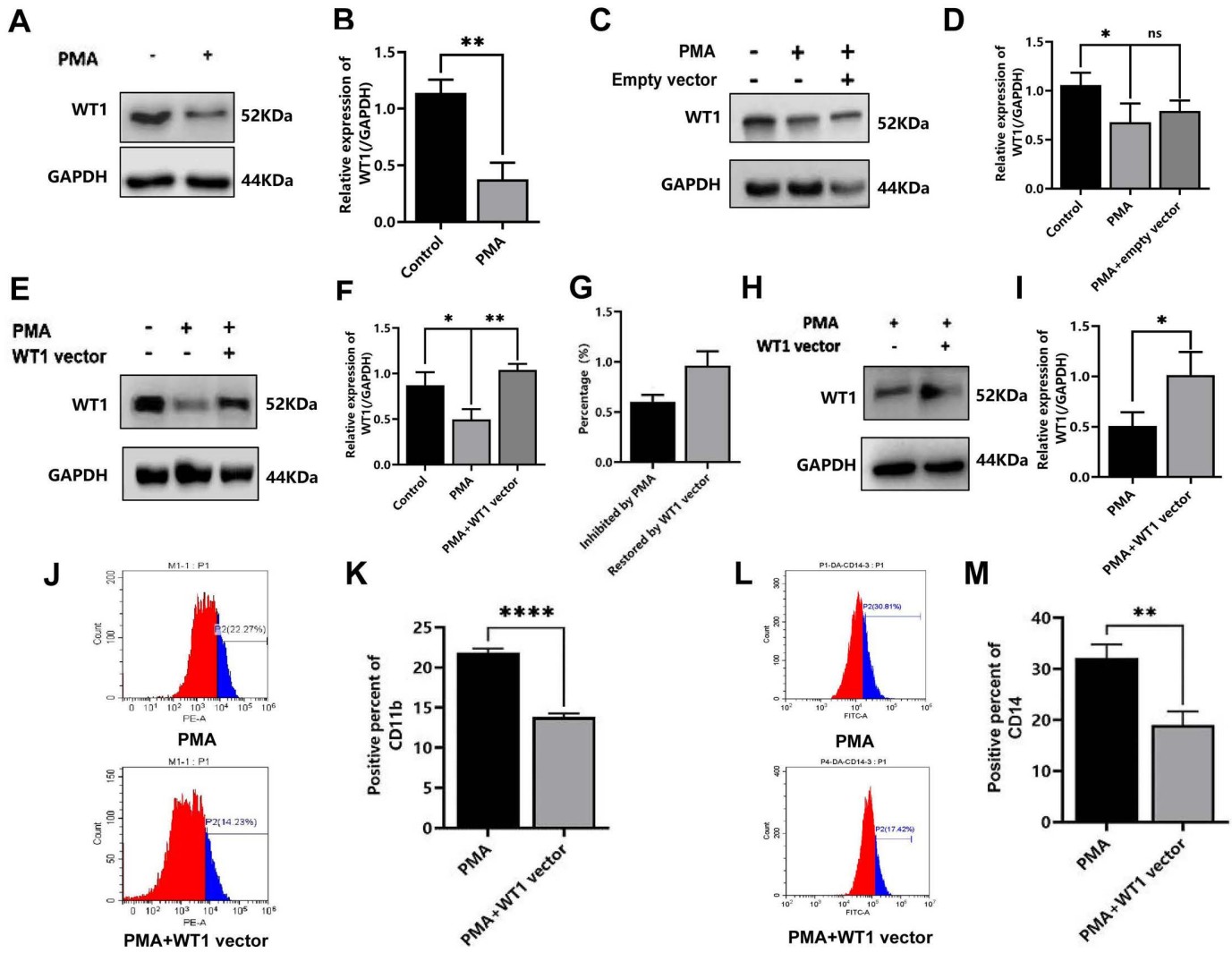

**Fig 3. Effect of WT1 gene overexpression on THP-1 cell differentiation induced by PMA.** Induced by PMA for 48 hours, the THP-1 cells were collected, the expression of WT1 cells was detected by western blotting assay, and the CD11b and CD14 of THP-1 cells was detected by flow cytometry. A. WT1 expression of THP-1 cells induced by PMA; B: Mean WT1 protein expression of THP-1 cells induced by PMA. C: WT1 expression of THP-1 cells after exposure to PMA or PMA+empty vector; D:Mean WT1 expression of THP-1 cells after exposure to PMA or PMA+empty vector; E: WT1 expression of THP-1 cells transfected with WT1 vector as compared with control group or PMA group; F: Mean relative expression of THP-1 cells transfected with WT1 vector as compared with control group or PMA group; G. Inhibiting percent of WT1 expression by PMA and the restoration percent of WT1 expression by PMA+WT1 vector to that of control group, which were based on the results of Fig 3E and 3F; H. The WT1 expression after exposure to PMA or PMA+WT1 vector by Western blot assay; I. Mean relative expression of WT1 in THP-1 cells after exposure to PMA or PMA+WT1 vector; J. The CD11b expression by flow cytometry assay; K. Expression of CD11b antigen after exposure to PMA or PMA+WT1 vector; L. The CD14 expression after exposure to PMA or PMA+WT1 vector by flow cytometry assay; M: Expression of CD14 antigens after exposure to PMA or PMA+WT1 vector. Each experiment was repeated three times. (* $P<0.05$, ** $P<0.01$, *** $P<0.001$).

miRNAs that were associated with WT1. The prediction results indicated that there were 17 miRNAs that could target and regulate WT1 (Fig 4A). To investigate whether differential changes occurred to the miRNAs in the THP-1 cells upon induction by PMA, RNA sequencing (RNA-seq) technology was used to identify differentially expressed genes and upregulated miRNAs, and a difference multiple >2 and a $P$-value <0.05 were taken as the boundaries. Through visual clustering analysis, it was found that there were significant increases or decreases in miRNAs both before and after PMA induction,

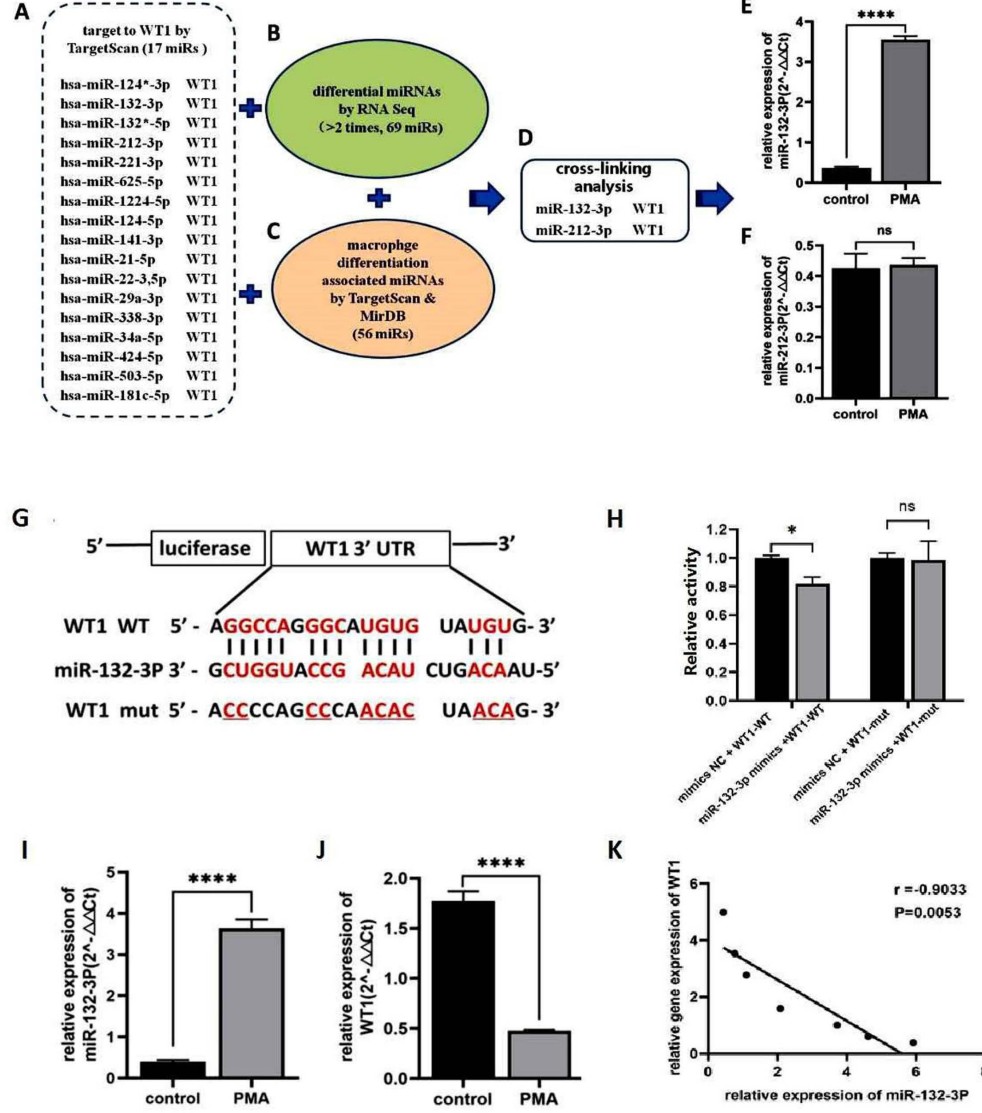

**Fig 4. Potential miRNA targeting WT1 and their relationship during committed differentiation towards macrophages.** TargetScan and miR-TarBase were used to predict potential upstream associated miRNAs of WT1. RNA seq technology to identify differentially altered genes (greater than 2 and a $P < 0.05$) as the boundary. TargetScan and MirDB databases were used to predict the miRNAs associated with monocyte/macrophage differentiation, and the intersection of the two databases was tried. Further validation of positively related miRNAs was demonstrated by qRT-PCR assay. In order to further verify whether miRNA-132-3p could bind WT1 3'UTR and modulate its promoter activity, a dual luciferase reporter assay was used. The relative expression of miRNAs was detected by qRT-PCR assay, and the correlation analysis between expression of miRNA-132-3p and WT1was indicated by Pearson analysis. A. TargetScan database predicts the possible miRNAs that target and bind to WT1; B. The number of differential miRNA species before and after differentiation and the number of macrophage differentiation-associated miRNA species analyzed by database; C. Target gene prediction, differential detection before and after differentiation, and database analysis macrophage-associated miRNA intersection analysis; D. miRNA-132-3p and miR-212-3p related to WT1 expression by intersection analysis; E: related expression of miRNA-132-3p before and after differentiation. F: miR-212-3p expression before and after differentiation; G: miRNA-132-3p targets WT1 3'UTR sequence prediction; H. Luciferase activity assay of miRNA-132-3p targeting WT1; I. The relative expression level of miRNA-132-3p in THP-1 cells before and after PMA induction; J. Relative expression levels of WT1 in THP-1 cells before and after PMA induction; K. Correlation analysis between miRNA-132-3p and WT1 expression. (ns: no significant, * $P < 0.05$, **** $P < 0.0001$).

and an MA plot also revealed the same increases or decreases of the miRNAs. A total of 69 miRNAs that were differentially expressed in THP-1 cells induced by PMA were identified (Fig 4B). The MirDB and TargetScan databases were also utilized to predict the miRNAs associated with macrophage differentiation, and upon analyzing the intersection of the screening results of the two databases, the results indicated that 56 miRNAs were associated with monocytes/macrophages (Fig 4C). Subsequently, the abovementioned miRNAs that could target WT1, the miRNAs associated with macrophage differentiation according to the database analysis, and the differentially expressed miRNAs before and after PMA induction were analyzed by intersection analysis, and these results indicated that only miRNA-132-3p and miR-212-3p were highly correlated with WT1 expression (Fig 4D). Further experiments revealed that, as compared with the control group, the expression of miRNA-132-3p in the THP-1 cells was significantly upregulated ($3.54 \pm 0.09$ vs. $0.36 \pm 0.03$, $t = 55.02$, $P < 0.0001$) (Fig 4E). However, by contrast, the expression of miR-212-3p remained almost unchanged before and after the induction of differentiation ($0.43 \pm 0.02$ vs. $0.43 \pm 0.04$, $t = 0.3462$, $P > 0.05$) (Fig 4F). Collectively, these results suggested that miRNA-132-3p may be an upstream miRNA associated with targeting the regulation of WT1, although further validation experiments need to be performed in this regard.

Furthermore, dual-luciferase reporter assay experiments were performed to detect the targeted binding of miRNA-132-3p to the 3'-UTR of WT1. The TargetScan database predicted that miRNA-132-3p may target binding to the 3'-UTR of WT1, although further validation of this point is required. In order to further verify whether miRNA-132-3p is targeted to the WT1 3'-UTR, a dual-luciferase reporter gene experiment was designed. First, the recombinant plasmids pmirGLO hWT1–3'-UTR-MUT and pmirGLO hWT1–3'-UTR-WT were constructed, and the WT1 wild-type and WT1 mutant plasmids constructed above were transfected into THP-1 cells, together with miRNA-132-3p mimics or NC mimics, respectively (Fig 4G). When co-transfected with miRNA-132-3p mimics, the luciferase activity of the pmirGLO hWT1–3'-UTR-WT group was lower compared with that of the co-transfection group of pmirGLO hWT1–3'-UTR-WT including NC mimics ($0.82 \pm 0.04$ vs. $1.00 \pm 0.01$; $t = 8.086$, $P < 0.05$, Fig 4H). This difference was found to be statistically significant. However, the luciferase activity of the WT1 mutant plasmid co-transfected with miRNA-132-3p mimics did not change significantly compared with NC mimics ($1.00 \pm 0.01$ vs. $0.98 \pm 0.13$, $t = 0.172$, $P > 0.05$) (Fig 4H). These results confirmed that miRNA-132-3p mimics could regulate the expression of WT1 by 'sponging' the 3'-UTR of WT1. In addition, the expression of miRNA-132-3p was found to be higher compared with that of the control group following exposure to PMA, and this difference was statistically significant ($3.63 \pm 0.21$ vs. $0.40 \pm 0.03$, $t = 25.77$, $P < 0.0001$) (Fig 4I). However, the expression level of WT1 was clearly downregulated compared with the control group ($0.47 \pm 0.01$ vs. $1.77 \pm 0.09$, $t = 23.13$, $P < 0.0001$) (Fig 4J). During the process of differentiation, miRNA-132-3p showed an increase in its expression level, whereas WT1 showed a decrease in expression level. During the 48 hour process of PMA induced differentiation of THP1 cells in vitro, 7 different time points were randomly selected, and qRT PCR was used to measure the changes in gene expression levels of miRNA 1323p and WT1, and analyze their correlation, and the results indicated that the miRNA-132-3p expression was found to be negatively correlated with WT1 expression, and this correlation was statistically significant ($r = -0.9033$, $P = 0.0053$) (Fig 4K).

## Rescue experiment of miRNA-132-3p targeting WT1 to regulate THP-1 cell differentiation

The above-mentioned results have been shown that downregulation of WT1 expression by PMA can induce differentiation of THP-1 cells, whereas miRNA-132-3p has a negative correlation with WT1. Therefore, a rescue experiment targeting miRNA-132-3p was designed to observe the differentiation of THP-1 cells via targeting WT1. The transfection efficiency of miRNA-132-3p mimics was firstly demonstrated by qRT-PCR (Fig 5A), and the transfection efficiency of WT1 vector was demonstrated by Western blotting experiments and the relative expression of WT1 was indicated as supplementary 1 Fig (Supplementary 1E, 1F Fig). Next, we conducted a reversal experiment of miR-132-3p mimetics for reducing WT1 expression. The results showed that, compared with the mimics NC group, miR-132-3p mimics could significantly reduce the expression level of WT1 (Fig 5B, 5C) ($t = 6.244$, $P = 0.0034$), while simultaneous transfection of miR-132-3p

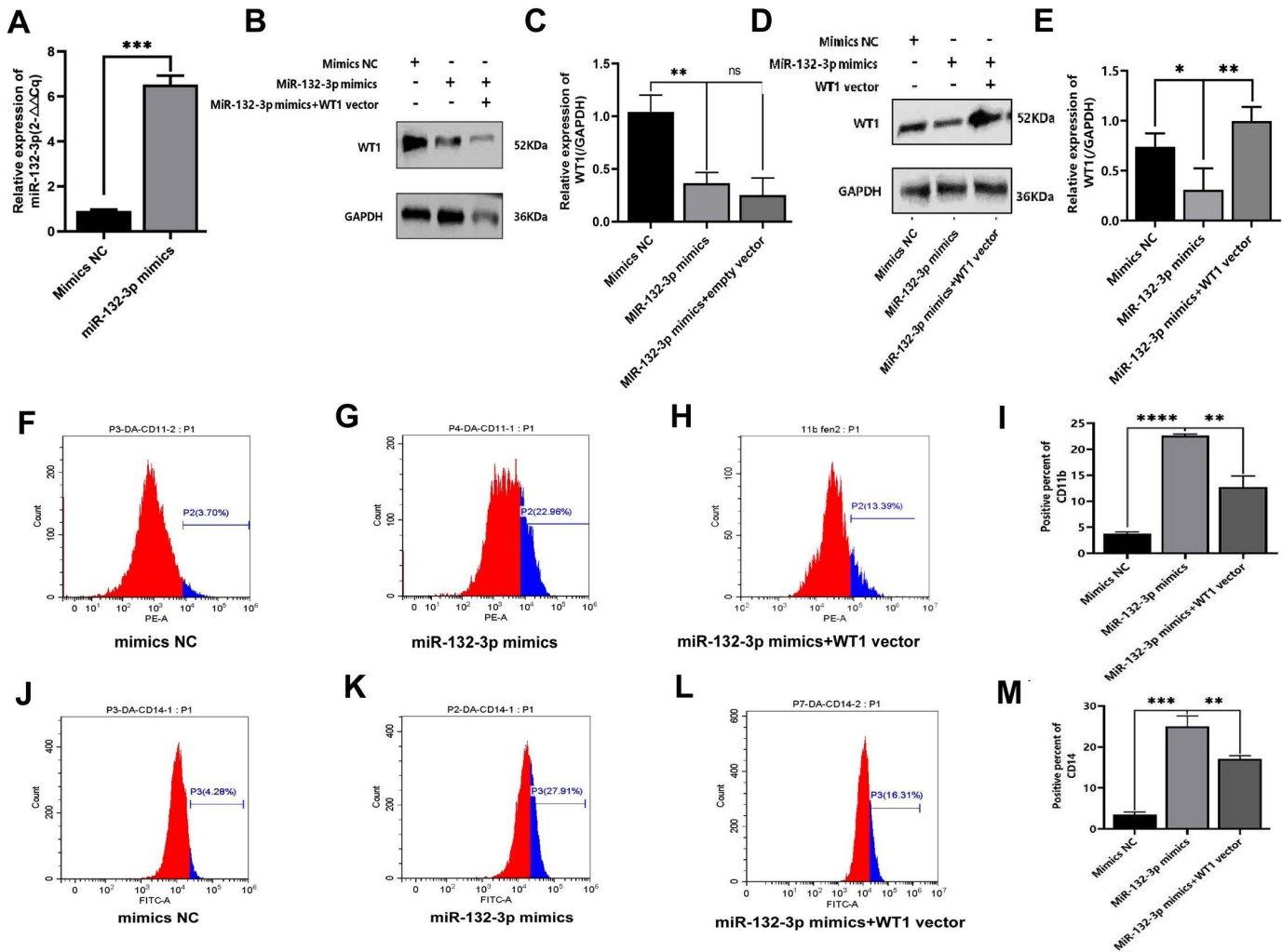

**Fig 5. Results of rescue experiments of miRNA-132-3p targeting WT1 to regulate the differentiation of THP-1 cells.** Rescue experiment about miRNA-132-3p to regulate differentiation of THP-1 cells by targeting WT1 was tried. WT1 protein expression was detected by Western blotting assay, the expression level of CD11b and CD14 differentiation antigen were detected by flow cytometry assay. A. The transfection efficiency of miRNA-132-3p mimics was detected by qRT-PCR; B. WT1 expression after exposure to mimics NC, miR-132-3p mimics or miR-132-3p mimics+empty vector; C.mean WT1 expression after exposure to mimics NC, miR-132-3p mimics or miR-132-3p mimics+empty vector; D. WT1 protein expression was detected by Western blot; E. Mean WT1 expression level; F. CD11b expression of mimics NC group by flow cytometry; G.CD11b expression of miR-132-3p mimics group by flow cytometry; H. CD11b expression of miR-132-3p mimics+WT1 vector group by flow cytometry; I.Mean expression level of CD11b differentiation antigen; J. CD14 expression of mimics NC group by flow cytometry; K.CD14 expression of miR-132-3p mimics group by flow cytometry; L.CD14 expression of miR-132-3p mimics+WT1 vector group by flow cytometry M. Mean expression levels of CD14 differentiation antigens. Each experiment was repeated three times.(** $P<0.01$, *** $P<0.001$).

mimics and empty vector had no significant effect on the independent regulation of WT1 by miR-132-3p mimics (t = 1.032, $P=0.3602$) (Fig 5B, 5C). Furthermore, compared with the mimics NC group, the WT1 expression in miRNA-132-3p mimics-transfected group was clearly downregulated (t = 2.945, $P=0.0422$). Moreover, the expression of WT1 in the miRNA-132-3p mimics+WT1 vector group was found to be higher as compared with that in the miRNA-132-3p mimics group, and the differences were statistically significant (t = 4.726, $P=0.0091$) (Fig 5D and 5E). Subsequently, the CD11b and CD14 antigens of the THP-1 cells were detected. These experiments showed that the expression of CD11b in the miRNA-132-3p mimics group was higher compared with that in the mimics NC group (22.65±0.27% vs. 3.78±0.32%,

t = 78.13, *P* < 0.001), and the expression of CD14 was also higher in the miRNA-132-3p mimics group (25.01 ± 2.52% vs. 3.55 ± 0.64%, t = 14.28, *P* = 0.0001). By contrast, the level of CD11b in the miRNA-132-3p mimics + WT1 vector group was lower compared with that in the miRNA-132-3p mimics group (12.77 ± 2.07% vs. 22.65 ± 0.27%, t = 8.212, *P* = 0.0012) (Figs 5F, 5G, 5H and 5I). The CD14 antigen was also downregulated in the miRNA-132-3p mimics + WT1 vector group compared with that of the miRNA-132-3p mimics group (17.15 ± 0.76% vs. 25.01 ± 2.52%, t = 5.170, *P* = 0.0067) (Figs 5J, 5K, 5L and 5M), and these differences were statistically significant. Taken together, these results suggested that overexpression of miRNA-132-3p led to a downregulation of the expression of WT1 through a negative regulation, thereby promoting the differentation of THP-1 cells towards macrophages.

## Upregulation of TGF-β1 expression by WT1 was detected during differentiation of THP-1 cells towards macrophages

The web tool GeneMANIA ([http://genemania.org/](http://genemania.org/)) and Alg</br>gen's online database were used to search or predict the association between WT1 and TGF-β1, and to investigate whether WT1 as a transcription factor can target the regulation of TGF-β1. Searching of the Alggen online database revealed that the TGF-β1 promoter region from -180 to + 1 bp contains possible binding sites for WT1, suggesting that WT1 may be closely associated with TGF-β1. Western blot analysis was first used to detect TGF-β1 expression of the THP-1 cells, and the results obtained showed that there was an increase in TGF-β1 expression following exposure to PMA (t = 8.069, *P* = 0.0013) (Fig 6A and 6B). To identify the role of WT1 in regulating its downstream target gene TGF-β1, the expression of TGF-β1 was observed upon transfecting with si-WT1, and the transfection efficiency of si-WT1 was demonstrated by the relative expression of WT1(Supplementary 1A, 1B Figs). These experiments revealed that, WT1 expression in the si-WT1-transfected group was downregulated compared with the control group (t = 4.974, *P* = 0.0076), however, there was no obvious difference between control group and si-WT1 scramble NC group (t = 0.7348, *P* = 0.5032) (Fig 6C and 6D). The expression of TGF-β1 was increased significantly in the si-WT1-transfected group as compared with that of control group (t = 11.16, P = 0.0004), and there was no significant difference between the control group and si-WT1 scramble NC group (t = 0.4209, *P* = 0.6955) (Fig 6C and 6E). As compared with the control group, the expression of CD11b was not affected significantly by s-WT1 scramble NC (t = 0.5512, *P* = 0.6109), however, the CD11b expression was upregulated in the si-WT1 group as compared with control group (t = 58.98, *P* < 0.0001) (Figs 6F, 6G, 6H and I). In addition, there was no obvious difference of CD14 expression between control group and si-WT1 scramble NC group (t = 0.7999, *P* = 0.4686), CD14 expression was significantly upregulated in the si-WT1 group as compared with control group (t = 50.49, *P* < 0.0001) (Figs 6J, 6K, 6L and 6M). Taken together, these results suggested that WT1 can negatively regulate TGF-β1 expression of its downstream target gene TGF-β1, and that downregulation of WT1 expression by siWT1 resulted in the upregulation of TGF-β1 expression.

## ChIP assay to detect WT1-targeted binding and regulation of TGF-β1 expression

To further verify whether WT1 targets the binding of TGF-β1, and regulates TGF-β1 expression, a ChIP assay experiment was designed. First, prediction analysis of the TGF-β1 promoter region for WT1 binding sites was performed, and a putative binding site in the region from -180 to + 1 bp was identified, as has been mentioned above (Fig 7A). Through performing database analyses and comparing the literature, the PCR amplification primer sequence of TGF-β1 was first determined (Fig 7B). Four groups were set up in this experiment, including the PMA + IgG group (negative control group), PMA + input group (positive control group), Control + anti-WT1 group, and PMA + anti-WT1 group. The ChIP-PCR experiments yielded a negative value for the PMA+ anti-WT1 group, whereas that for the Control + anti-WT1 group was positive (Fig 7C). Taken together, these results confirmed that WT1 can target both the binding and regulation of TGF-β1 expression.

Rescue experiment to investigate the role of miRNA-132-3p in inducing THP-1 cell differentiation via modulating the expression of TGF-β1. Subsequently, experiments were designed comprising the following experimental groups, namely

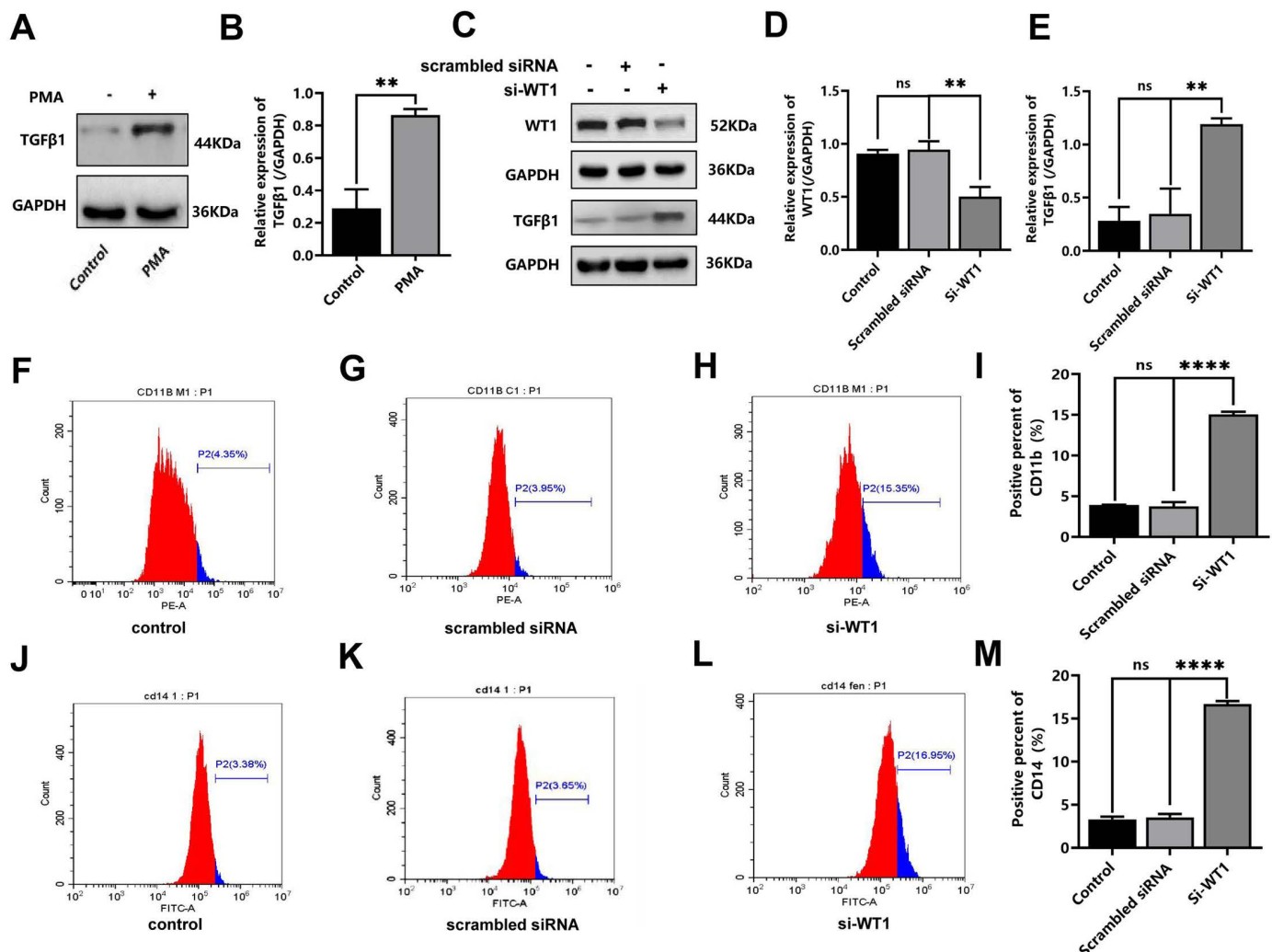

**Fig 6. Targeting regulation of TGF-β1 expression by WT1 on committed differentiation of monocytes/macrophages.** WT1 protein expression was detected by Western blotting assay, and the expression level of CD11b and CD14 differentiation antigen were detected by flow cytometry assay. A. Western blot was used to detect the protein level of TGF-β1 in THP-1 cells after PMA treatment; B. Changes in the mean level of TGF-β1 protein in THP-1 cells after PMA induction; C. The expression of WT1 or TGF-β1 protein in THP-1 cells transfected with si-WT1 scramble NC or si-WT1; D. Mean WT1 protein expression level in THP-1 cells transfected with si-WT1 scramble NC or si-WT1; E. Mean TGF-β1 protein expression in THP-1 cells transfected with si-WT1 scramble NC or si-WT1; F. Expression of CD11b antigens of control group by flow cytometry; G. Expression of CD11b antigens of si-WT1 scramble NC group by flow cytometry; H. Expression of CD11b antigens of si-WT1 group by flow cytometry; I. Mean CD11b antigens of above-mentioned indicated group; J. Expression of CD14 antigens of control group by flow cytometry; K. Expression of CD14 antigens of si-WT1 scramble NC group by flow cytometry; L. Expression of CD14 antigens of si-WT1 group by flow cytometry; M. Mean CD14 antigens of above-mentioned indicated group;. Each experiment was repeated three times, (* $P < 0.05$, *** $P < 0.001$, **** $P < 0.0001$).

the mimics-NC group, miRNA-132-3p mimics group and miRNA-132-3p mimics+si-TGF-β1 group. Western blotting assay was used to detect the protein expression of TGF-β1, and the transfection efficiency of si-TGF-β1 was demonstrated by the relative expression of WT1(Supplementary 1C, 1D Figs). To confirm the role of TGF-β1 in mediating regulation of cell differentiation by miR-132-3p mimics, we conducted a rescue experiment of si-TGF-β1. Firstly, in the transfection experiment of miR-132-3p mimics, it was found that the expression level of TGF-β1 in the miR-132-3p mimics transfection group was significantly higher than that of the mimics NC control group (t = 14.13, P = 0.0001). When miR-132-3p

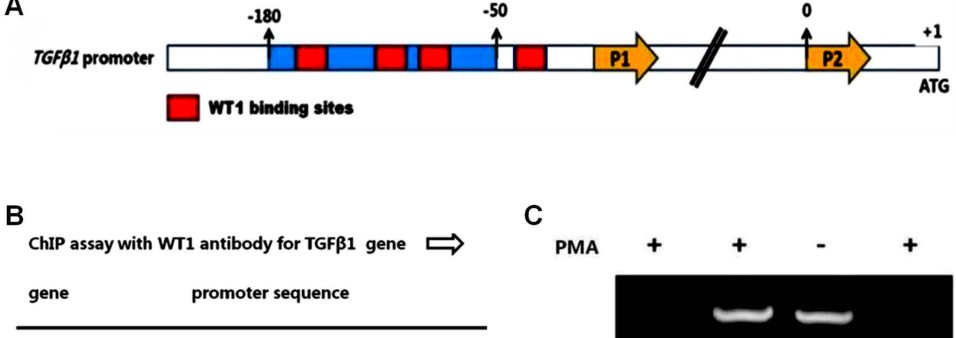

**Fig 7. Chromatin immunoprecipitation (ChIP) assay for WT1 targeting and regulation of TGF- β1 expression.** To further verify whether WT1 targets binding and regulates expression of TGF-β1, Chromosome immunoprecipitation (ChIP) assay for targeting regulation of TGF-β1 by WT1. A. Prediction of the binding region of WT1-targeted TGF-β1 promoter; B. Primer sequence design for ChIP assay of TGF-β1; C. ChIP assay results of WT1 targeting TGF-β1.

mimics were transfected simultaneously with si-TGF-β1 scramble NC, there was no significant change of the expression of TGF-β1 between these two groups (0.2772, P = 0.7954) (Figs 8A and 8B). By contrast, the expression of TGF-β1 in the miRNA-132-3p mimics+si-TGF-β1 transfected group was lower as compared with that of the miRNA-132-3p mimics only group (t = 3.866, P = 0.0181) (Figs 8C and 8D). Subsequently, the differentiation antigens of CD11b and CD14 were detected by flow cytometry assay, and the results revealed that the expression of both CD11b and CD14 in the miRNA-132-3p mimics group was upregulated compared with that in the mimics NC group (t = 41.23, P < 0.0001; t = 22.96, P < 0.0001, respectively). In addition, the expression of the CD11b antigens in the miRNA-132-3p mimics + si-TGF-β1-transfected group was downregulated compared with that in the miRNA-132-3p mimics group, and the differences were found to be statistically significant (20.04 ± 0.66 vs 8.49 ± 0.95, t = 17.18, P < 0.0001) (Figs 8E, 8F, 8G and 8H). The expression of the CD14 antigens in the miRNA-132-3p mimics + si-TGF-β1-transfected group was downregulated compared with that in the miRNA-132-3p mimics group (Figs 8I, 8J, 8K and 8L), and the differences were found to be statistically significant (24.31 ± 1.28 vs 9.13 ± 2.50, t = 9.327, P = 0.0007, respectively). Taken together, these results indicated that miRNA-132-3p is able to positively regulate TGF-β1 expression, and an elevated level of miRNA-132-3p expression upregulates TGF-β1 expression, thereby promoting THP-1 cell-directed macrophage differentiation, whereas siTGF-β1 causes a reversal of this role of miRNA-132-3p in modulating expression of CD11b and CD14 antigens.

The miRNA-132-3p/WT1/TGF-β1 signaling pathway is involved in inducing differentiation of THP-1 cells towards macrophages induced by PMA. Considering all the results obtained thus far, it was possible to conclude that PMA can induce the committed differentiation of THP-1 cells towards macrophages. During this committed differentiation process, miRNA-132-3p expression is upregulated, WT1 expression is reduced, and TGF-β1 expression is also upregulated. Moreover, these three genes have been demonstrated to be involved in this committed macrophage differentiation of leukemia cells. Through target gene prediction and the dual-luciferase reporter gene assay experiments, it was confirmed that miRNA-132-3p regulates the level of promoter activity of the WT1 gene through targeting its 3'-UTR region, thereby regulating the committed macrophage differentiation of leukemia cells. Subsequently, it was also demonstrated that miRNA-132-3p can regulate TGF-β1 by targeting WT1, thereby promoting the differentiation of THP-1 cells towards macrophages. In conclusion, the miRNA-132-3p/WT1/TGF-β1 axis has been shown to contribute to the polarization of THP-1 cells towards macrophages (Fig 9).

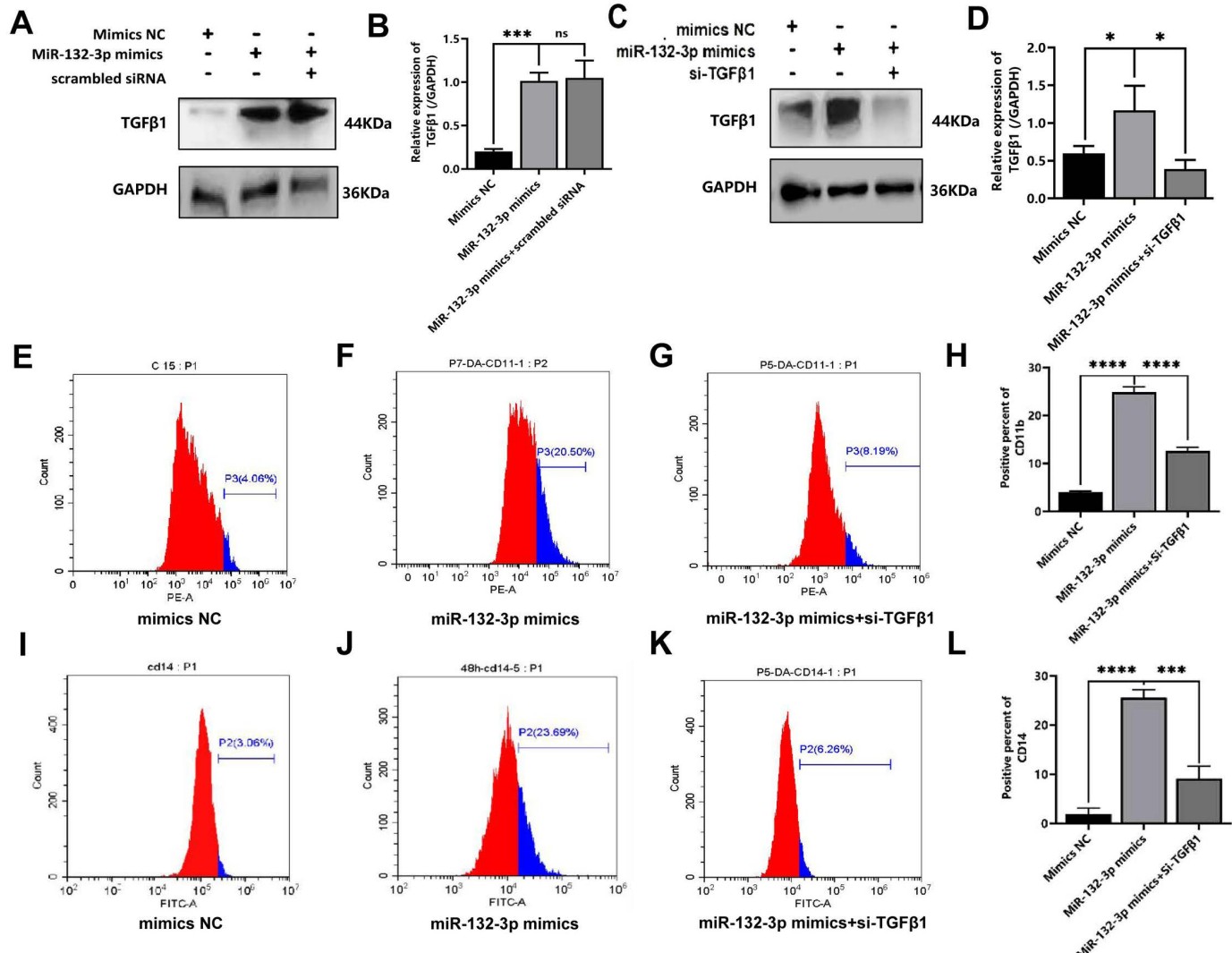

**Fig 8. Rescue assay results of miRNA-132-3p targeting TGF- β1 to induce differentiation of THP-1 cells. miRNA-132-3p mimics-NC group, miRNA-132-3p mimics group, and miRNA-132-3p mimics+si-TGF-β1 group three groups were used to try rescue assay about miRNA-132-3p modulating expression of TGF-β1.** Western blotting assay was used for detecting expression of TGF-β1 protein. A. TGF-β1 expression in the indicated mimics NC, miR-132-3p mimics or miR-132-3p mimics+si-TGF-β1 scramble NC groups by Western blot assay; B.Mean expression of TGF-β1 protein in the above indicated groups; C. TGF-β1 expression in the indicated mimics NC, miR-132-3p mimics or miR-132-3p mimics+si-TGF-β1 groups by Western blot assay; D. Mean expression of TGF-β1 protein in the above indicated groups; E. Expression of CD11b antigens of mimics NC group by flow cytometry;.F. Expression of CD11b antigens of miR-132-3p mimics group by flow cytometry; G. Expression of CD11b antigens of miR-132-3p mimics+WT1 vector group by flow cytometry; H.Mean expression of CD11b in the indicated groups; I. Expression of CD14 antigens of mimics NC group by flow cytometry;.J. Expression of CD14 antigens of miR-132-3p mimics group by flow cytometry; K. Expression of CD14 antigens of miR-132-3p mimics+WT1 vector group by flow cytometry; L.Mean expression of CD14 antigens in the indicated groups;.Each experiment was repeated three times, (* $P < 0.05$, ** $P < 0.01$).

## Discussion

AML is a malignant clonal disease of hematopoietic stem cells that blocks the differentiation of myeloid cells [46,47]. In addition to conventional drug-targeted therapy and chemotherapy, interestingly, it has been demonstrated that induction differentiation therapy is an ideal therapeutic method for the treatment of leukemia, and has been successfully used in the

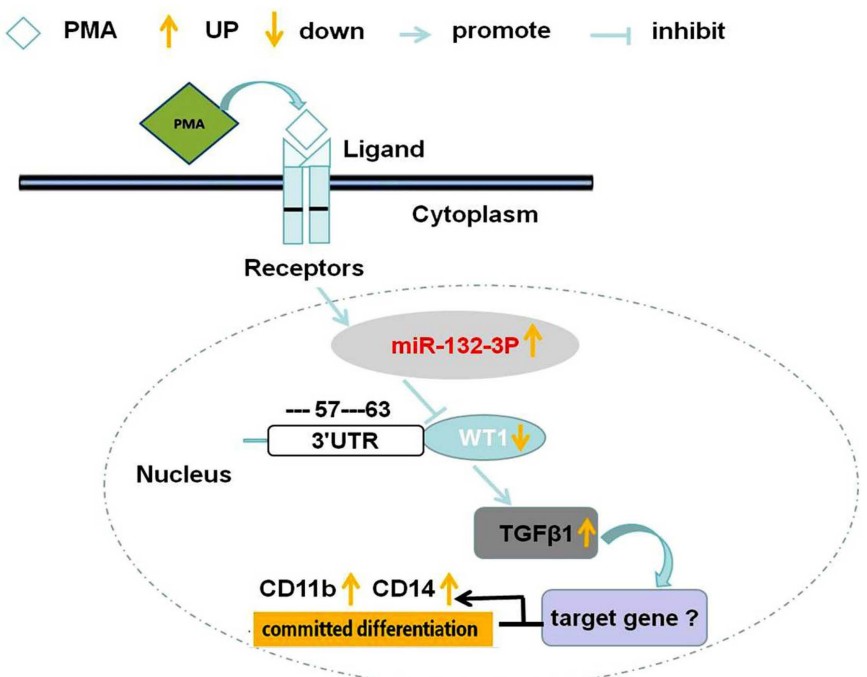

**Fig 9. Schematic of the signaling pathway of the miRNA-132-3p/WT1/TGF- β1 axis contributed to inducing differentiation of THP-1 cells into macrophages.** In THP-1 cells, PMA could induce the upregulation of miRNA-132-3p, the upregulation of miRNA-132-3p results in the downregulation of WT1 by sponge action with binding on its 3'UTR, the down regulation of WT1 contributes to the upregulation of TGF-β1, then the polarization of macrophages is induced, which is indicated by the upregulation of CD11b and CD14 expression. Therefore, the miRNA-132-3p/WT1/TGF-β1 axis is involved in the committed differentiation of THP-1 leukemia cells into macrophages.

clinic for the treatment of patients with APL [48–52]. However, to date, leukemia cells have only been successfully induced to differentiate into granulocytes, and no significant breakthroughs have been achieved in inducing differentiation towards monocytes and macrophages. The quest to unravel the underlying molecular mechanisms of leukemia differentiation disorders is of crucial importance, as it provides an important basis for finding targeted therapeutic targets. However, the selection of the research entry point is directly related to whether the key signaling pathways of leukemia cell differentiation disorders can be uncovered, especially the discovery of key regulatory molecules or important signaling pathways that are associated with the differentiation of the leukemia cells.

According to previous studies, WT1, as an effective transcription regulatory factor, has been shown to be closely associated with the proliferation, apoptosis, differentiation and survival of normal or tumor cells [53–55], or WT1 is an important factor associated with the occurrence and development of leukemia [56]. However, although WT1 is well established as one of the oncogenes closely associated with tumors or leukemia, to date, few studies have reported on its important role in the targeted differentiation of leukemia. Due to its involvement in the proliferation and differentiation processes of tumor cells, however, it may be speculated that it exerts its role through participating in a certain signaling pathway.

Accordingly, we hypothesized that WT1 may be involved in the committed differentiation of leukemia cells. In the preliminary experiments, differential gene expression between THP-1 cells and normal peripheral blood monocytes was analyzed through RNA-seq. The results obtained confirmed that WT1 is abnormally overexpressed in leukemia cells, and enrichment analysis revealed that it is associated with the process of cell differentiation. In order to further reveal the role of WT1 in the committed macrophage differentiation of leukemia cells, in the present study, THP-1 cells were induced by PMA to establish a representative cell model for macrophage-committed differentiation, as described previously [53,54]. It

was found that PMA could induce the committed differentiation of THP-1 leukemia cells towards macrophages, as evaluated by the upregulation of CD11b and CD14 antigens, and the expression of WT1 protein was gradually decreased. The si-WT1 transfection assay also demonstrated the downregualtion of WT1 expression in committed differentiation of THP1 cells. However, more research works was required to determine how WT1 participates in the 'switch' from THP-1 cells into macrophages.

miRNAs are endogenous noncoding small RNAs that usually target the 3'-UTRs of specific target mRNAs to regulate the expression of target genes [57], resulting in the regulation of cell differentiation or apoptosis [58], and a number of different studies have demonstrated that certain miRNAs may function as potential biomarkers for cancer diagnosis and treatment [59]. Usually, miRNAs exert their role through a 'sponging' action to modulate downstream genes. This serves the purpose of regulating the expression of downstream oncogenes or tumor suppressors, such as targeting WT1 to regulate the proliferation of tumor cells [60,61]. To confirm the hypothesis that some miRNAs may involve in differentiation of THP1 cells by targeting WT1, RNA-seq was used as the technology to screen and analyze differentially expressed miRNAs both before and after PMA induction. On the other hand, based on the fact that miRNAs often exhibit a negative correlation with the expression levels of downstream target gene mRNAs, we focused on analyzing the differentially expressed miRNAs that were upregulated following exposure to PMA. To reveal the specific role of WT1 in the committed differentiation, its upstream microRNAs were predicted using the Targetscan and mirDB databases. Furthermore, literature research results on the targeted regulation of WT1 were manually searched using the PubMed/MEDLINE database. After performing an intersection analysis of these three related investigations, together with performing an additional RT-qPCR analysis for identification purposes, miRNA-132-3p emerged as the miRNA most likely to regulate WT1 expression. Therefore, miRNA-132-3p was identified as the potential upstream miRNA targeting WT1, which was predicted using the Targetscan database. Dual luciferase reporter gene assay further confirmed that WT1 is truly a downstream gene of miRNA-132-3p, and the application of miRNA-132-3p mimics resulted in a decrease in WT1 expression, accompanied by an upregulation of the CD11b and CD14 antigens. The experimental results of interference and rescue have demonstrated that miRNA-132-3p participates in PMA induced macrophage polarization of THP1 cells by regulating the expression level of WT1. The preliminary research results suggest that miRNA 1323p may be a potential tumor suppressor miRNA, with similar functions to other tumor suppressor miRNAs in blocking and mitigating leukemia progression [62].

After having confirmed the identity of the upstream miRNA of WT1, subsequently, whether WT1 has a role in targeting downstream target genes became the focus of our further research. Based on a literature analysis, as an important member of the TGF family, TGF-β1 was identified not only as a participant in cell proliferation, differentiation, and apoptosis, but it also suppresses the proliferation of tumor cells, or induces differentiation of leukemia cells [63,64]. Therefore, we hypothesized that WT1 may regulate the differentiation of THP-1 cells into macrophages via regulating the TGF-β1. WT1 may be a possible transcription factor targeting the promoter of TGF-β1 was first identified through database prediction analysis. Subsequently, a ChIP assay was performed to establish whether WT1 could interact with the promoter of TGF-β1, and whether WT1 could negatively regulate the expression of TGF-β1 to promote the differentiation of THP-1 cells into macrophages. Furthermore, we sought to investigate whether miRNA-132-3p could affect the differentiation of THP-1 cells via regulating TGF-β1 expression. Taken together, the results of these experiments indicated that miRNA-132-3p was able to positively regulate TGF-β1 expression, and the interference or rescue experiment demonstrated that miRNA-132-3p indeed exerts its role through regulating the expression of TGF-β1.

Although it has been preliminarily shown that miRNA-132-3p, WT1 and TGF-β1 have important roles in the PMA-induced differentiation of THP-1 cells into macrophages by way of upstream or downstream regulation, further studies are required in order to fully delineate the underlying mechanism, and to reveal the significance of their regulatory associations. For example, the upstream genes or transcription regulatory proteins that regulate the expression of miRNA-132-3p need to be identified, and the manner in which TGF-β1 regulates its downstream target genes still needs to be explored. Secondly, given that it is typically encountered as the downstream gene of the TGF-β1 signaling pathway [65,66], the role

of the Smad family in inducing differentiation of leukemia cells into macrophages needs to be elucidated. In addition, in order to clarify the changes observed in these three genes, and the significance of the signaling axis that they regulate, the required in vivo animal studies and the detection of primary leukemia cells in clinical patients will be important aspects of our subsequent experiments. In spite of this, the present work, which was mainly focused on THP-1 cells, did achieve the hoped-for results. Furthermore, to corroborate the work done with the THP-1 cells, the other more leukemia cell lines, or primary leukemia cells of patients with different types will be utilized in future studies to investigate whether the miR-132-3p/WT1/TGF-β1 signaling axis exerts a similar role in the committed differentiation into macrophages, which can be applied to leukemic cells more in general.

In conclusion, although higher expression of WT1 has been demonstrated in patients with leukemia [67,68], however, the exact role of WT1 in the development of leukemia is still not fully understood. The complexity of its functions, multiple isomers, and the lack of ideal research models all pose challenges to further clarify its functional characteristics [69,70]. So far, many studies on the mechanism of WT1 have focused on its function as a transcription factors, and recent works from AML large-scale genome research has revealed a new role of WT1 in epigenetic regulation [71]. On the other hand, just due to the abnormal expression of WT1, some scholars have confirmed that it can serve as an indicator for minimal residual disease (MRD) monitoring and prognosis judgment [72]. The present study has demonstrated, to the best of our knowledge for the first time, that the miRNA-132-3p/WT1/TGF-β1 signaling axis is an important pathway that is involved in the committed differentiation of THP-1 leukemia cells into macrophages, providing a good theoretical basis underpinning future in-depth research on the clinical significance of this signaling axis. The importance of the miRNA-132-3p/WT1/TGF-β1 signaling axis as a prognostic indicator for patients with leukemia needs to be validated through further clinical studies, and the potential value of targeted intervention therapy needs to be validated at the animal level first. Inducers or inhibitors of this signaling pathway also have the potential to become targeted intervention drugs. The further in-depth research on more cell lines, as well as the results of clinical specimens detection and animal experiments, will enhance the theoretical and practical value of this signaling pathway.

## Supporting information

**S1 Fig. The transfection efficiency of si-WT1, si-TGF-β1 and WT1 vector detected by the relative expression of their proteins.** A. The WT1 expression by Western blotting assay in THP-1 cells transfected with siNC and si-WT1; B. The mean relative expression of WT1 in THP-1 cells transfected with siNC and si-WT1; C. The TGF-β1 expression by Western blotting assay in THP-1 cells transfected with siNC and si-TGF-β1; D. The mean relative expression of TGF-β1 in THP-1 cells transfected with siNC and si-TGF-β1; E. The WT1 expression by Western blotting assay in THP-1 cells transfected with empty vector and WT1 vector; F. The mean relative expression of WT1 in THP-1 cells transfected with empty vector and WT1 vector.
(TIF)

## Author contributions

**Conceptualization:** Chaozhe Wang, Kehong Bi, Guosheng Jiang.

**Data curation:** Chaozhe Wang, Ruijing Sun, Xiaolin Sun.

**Formal analysis:** Xidi Wang, Yunhua Wu, Ruijing Sun.

**Investigation:** Zhimin Wang, Danfeng Zhang, Yunhua Wu.

**Methodology:** Danfeng Zhang, Xidi Wang, Ruijing Sun, Xiaolin Sun.

**Project administration:** Qing Li, Kehong Bi, Guosheng Jiang.

**Software:** Zhimin Wang, Danfeng Zhang, Xidi Wang, Yunhua Wu, Ruijing Sun, Xiaolin Sun.

**Supervision:** Qing Li, Kehong Bi, Guosheng Jiang.

**Validation:** Yunhua Wu, Qing Li, Kehong Bi, Guosheng Jiang.

**Visualization:** Qing Li, Kehong Bi, Guosheng Jiang.

**Writing – original draft:** Zhimin Wang, Chaozhe Wang.

**Writing – review & editing:** Kehong Bi, Guosheng Jiang.

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
