## [Decision Letter · Decision Letter 0]

30 Jul 2024

PONE-D-24-22408The up-regulation of TGF-β1 by miRNA-132-3p/WT1 is involved in inducing leukemia cells to differentiate into macrophagesPLOS ONE

Dear Dr. Wang,

Thank you for submitting your manuscript to PLOS ONE. After careful consideration, we feel that it has merit but does not fully meet PLOS ONE’s publication criteria as it currently stands. Therefore, we invite you to submit a revised version of the manuscript that addresses the points raised during the review process.Please note that the manuscript can be taken again into consideration only if you address in detail all the comments from Reviewer #2, an expert in the field.

We look forward to receiving your revised manuscript.

Kind regards,

Francesco Bertolini, MD, PhD

Academic Editor

PLOS ONE

4. PLOS requires an ORCID iD for the corresponding author in Editorial Manager on papers submitted after December 6th, 2016. Please ensure that you have an ORCID iD and that it is validated in Editorial Manager. To do this, go to ‘Update my Information’ (in the upper left-hand corner of the main menu), and click on the Fetch/Validate link next to the ORCID field. This will take you to the ORCID site and allow you to create a new iD or authenticate a pre-existing iD in Editorial Manager. Please see the following video for instructions on linking an ORCID iD to your Editorial Manager account: https://www.youtube.com/watch?v=_xcclfuvtxQ.

5. lease include captions for your Supporting Information files at the end of your manuscript, and update any in-text citations to match accordingly. Please see our Supporting Information guidelines for more information: http://journals.plos.org/plosone/s/supporting-information. 

Additional Editor Comments (if provided):

Reviewers' comments:

Reviewer's Responses to Questions

**Comments to the Author**

1. Is the manuscript technically sound, and do the data support the conclusions?

Reviewer #1: Yes

Reviewer #2: Partly

2. Has the statistical analysis been performed appropriately and rigorously?

Reviewer #1: Yes

Reviewer #2: Yes

3. Have the authors made all data underlying the findings in their manuscript fully available?

Reviewer #1: Yes

Reviewer #2: Yes

4. Is the manuscript presented in an intelligible fashion and written in standard English?

Reviewer #1: Yes

Reviewer #2: No

5. Review Comments to the Author

Reviewer #1: Zhimin and colleagues presented a very nice work in which they demonstrated that a specific microRNA 132-3p regulates WT1 gene. Moreover, they were also able to correlate this interaction with TGFbeta.

This reviewer has no concern, the experiments are logical, presented in a clear and readble way.

Minor concern:

1. figure are not always readble, they should be improved in their quality

2. Supplementary Figure is written in a wrong way in all the paper, is written "supplimentary"

Reviewer #2: The manuscript from Wang Zhimin et al. aims to address the role of WT1 as potential player in the differentiation of leukemic cells into macrophages and to dissect the pathways involved. We agree with the authors that, although WT1 is frequently found mutated and/or overexpressed in a number of cancers, its role in the leukemic process still need to be clarified and that exploring cellular differentiation as a possible therapeutic strategy is of general interest. Nonetheless, I think that the paper is not suitable for publication in its present form.

Most key experiments lack the correct controls and, therefore, the results are difficult to judge and it is impossible to draw solid conclusions. In particular: i) all rescue experiments, both with overexpression vectors and siRNA targets, do not show the results obtained with the controls (i.e. empty vector, scrambled siRNA). ii) In the differentiation experiments mediated by PMA, the basal level of expression of the differentiation markers in absence of PMA treatment should always be shown, in order to be sure of the PMA efficacy in each experiment.

In most figures, the results of western blot analysis are shown as independent panels, each target in one panel. This suggest that each panel is the result of an independent experiment, making it impossible to define a direct link and effect among the different players (i.e. Fig 6C and E).

Some experiments and plots are not well explained and detailed. For example, for Figure 4K how is the correlation analysis of expression performed? Which data are plotted, how measured?

There is no indication of how many times experiments have been reproduced.

Moreover, as also mentioned by the authors in the Discussion, I think that, in order to define if the results of this study are not model-dependent and can be applied to leukemic cells more in general, the results should be reproduced in at least another independent cell line.

Introduction and Discussion are largely overlapping, I think that the authors should try not to be too repetitive. Even some experiments and how they are described in the Results are redundant (i.e. Fig. 3 A-B and Fig. 3F-G show the same results of Fig. 3C-D).

Finally, in general, English language, typos and paragraph formatting should be checked.

Figure 2A and B, should be replaced with panels of better quality and higher magnification (the cell types are difficult to distinguish in the provided micrographs).

6. PLOS authors have the option to publish the peer review history of their article (what does this mean? ). If published, this will include your full peer review and any attached files.

**Do you want your identity to be public for this peer review?** For information about this choice, including consent withdrawal, please see our Privacy Policy .

Reviewer #1: No

Reviewer #2: No

---

## [Author Response · Author response to Decision Letter 1]

22 Jan 2025

Dear editor

Thank you and reviewers very much for your valuable suggestions and opinions once again.These suggestions are not only beneficial for improving the quality of the present manuscript, but also have a very helpful effect on our future related research and paper writing.Based on your suggestions, we have carefully reviewed and made corresponding modifications and supplements.

Comments to the Author

1. Is the manuscript technically sound, and do the data support the conclusions?

Reviewer #1: Yes

Reviewer #2: Partly

2. Has the statistical analysis been performed appropriately and rigorously?

Reviewer #1: Yes

Reviewer #2: Yes

3. Have the authors made all data underlying the findings in their manuscript fully available?

Reviewer #1: Yes

Reviewer #2: Yes

4. Is the manuscript presented in an intelligible fashion and written in standard English?

Reviewer #1: Yes

Reviewer #2: No

5. Review Comments to the Author

Please use the space provided to explain your answers to the questions above. You may also include additional comments for the author, including concerns about dual publication, research ethics, or publication ethics. (Please upload your review as an attachment if it exceeds 20,000 characters)。

Reviewer #1:

Question 1: Zhimin and colleagues presented a very nice work in which they demonstrated that a specific microRNA 132-3p regulates WT1 gene. Moreover, they were also able to correlate this interaction with TGFbeta.This reviewer has no concern, the experiments are logical, presented in a clear and readble way.

Answer: Thank you very much for your support and recognition of our work. We will focus on carefully discussing and revising the minor concerns you have raised. I also hope to provide opinions and suggestions.

Question 2:. figure are not always readble, they should be improved in their quality

Answer:Thank you very much for your suggestions on the quality of the result images. After careful verification, we have made modifications and improvements to the resolution and clarity of some images based on your suggestions, and replaced some of the result images. Please refer to the resubmitted result chart and provide suggestions.

Question 3. Supplementary Figure is written in a wrong way in all the paper, is written "supplimentary".

Answer: So sorry for the wrong spelling about the word “supplimentary”, and it is changed to the right word about “supplementary” as what you suggested. Please check it in the new manuscript.

Reviewer #2:

Question 1:The manuscript from Wang Zhimin et al. aims to address the role of WT1 as potential player in the differentiation of leukemic cells into macrophages and to dissect the pathways involved. We agree with the authors that, although WT1 is frequently found mutated and/or overexpressed in a number of cancers, its role in the leukemic process still need to be clarified and that exploring cellular differentiation as a possible therapeutic strategy is of general interest.

Answer: Thank you very much for acknowledging and recognizing the significance of our research paper. Leukemia induced differentiation therapy is one of the most ideal treatment methods, with the advantage of selectively targeting leukemia cells without significant side effects on normal hematopoiesis and immunity. Therefore, it is necessary to further explore the molecular mechanisms of committed differentiation in leukemia and provide a basis for targeted therapy.

Question 2:Nonetheless, I think that the paper is not suitable for publication in its present form. Most key experiments lack the correct controls and, therefore, the results are difficult to judge and it is impossible to draw solid conclusions.In particular: i) all rescue experiments, both with overexpression vectors and siRNA targets, do not show the results obtained with the controls (i.e. empty vector, scrambled siRNA).

Answer: Thank you very much for your suggestions and opinions on the rescue experiment. The setting of a negative control group does indeed affect whether clear conclusions can be obtained. Based on your suggestion, we have conducted a thorough review and discussion, and provided necessary explanations and supplements accordingly. For example, we have added the control group of empty vectors in Figure 3C and Figure 5B respectively. In Figure 6C and 8A,we added si-WT1 scramble NC and si-TGF β 1 scramble NC control groups respectively , and replaced the original negative control. Of course, the negative control group we originally used was the commonly used si-NC, which is also the negative control used in the vast majority of current articles. Of course, Scramble NC is more ideal as a negative control.

Question 3: ii) In the differentiation experiments mediated by PMA, the basal level of expression of the differentiation markers in absence of PMA treatment should always be shown, in order to be sure of the PMA efficacy in each experiment.

Answer: Thank you very much for this important suggestion. The expression level of differentiation antigens in the non-drug negative control group is indeed very important. However, we need to provide special clarification on this point. In Figure 2, we first demonstrate the changes in differentiation antigen expression before and after PMA treatment, including the basal level of expression of the differentiation markers in absence of PMA treatment. Therefore, we did not repeat this experiment repeatedly in the following section of the results.Please confirm if our explanation is reasonable.

Question 4: In most figures, the results of western blot analysis are shown as independent panels, each target in one panel. This suggest that each panel is the result of an independent experiment, making it impossible to define a direct link and effect among the different players (i.e. Fig 6C and E).

Answer: Thank you for this important suggestion. Based on your suggestion, we have re-performed the Western blot assay about some indicated proteins,for example, the original 6C and E images, and the new results are as shown in the new Figure 6C. However, we need to provide an explanation and clarification for the Western blot analysis experiment. The original Western blotting results mainly relied on internal reference protein as quality control, such as GAPDH or β-actin and demonstrated the average relative density according to the usual three repeated experiments.

Question 5�Some experiments and plots are not well explained and detailed. For example, for Figure 4K how is the correlation analysis of expression performed? Which data are plotted, how measured?

Answer: I am very sorry for the omission in the explanation and interpretation of some of the result graphs or plots. Based on your suggestion, we have made corresponding supplements and explanations in the materials and methods section, as well as in the corresponding results section. Please review the corresponding Figure 4K section of the new manuscript.

Question 6:There is no indication of how many times experiments have been reproduced.

Answer: Thank you for the reminder and suggestion regarding the number of repetitions in the experiment. We originally only conducted three repeated experiments in the statistical methods section, but now we have also provided annotations and explanations in the Figure legend section.

Question 7: Moreover, as also mentioned by the authors in the Discussion, I think that, in order to define if the results of this study are not model-dependent and can be applied to leukemic cells more in general, the results should be reproduced in at least another independent cell line.

Answer: Thank you for your suggestion and I strongly agree with this viewpoint. More cell lines, and even more primary leukemia cell studies, will help confirm the universality of this signaling pathway, which is also a suggestion we actively raised in our discussion. However, leukemia is a highly heterogeneous disease, and different cell lines of the same type may also exhibit heterogeneity. Patients with the same subtype of leukemia may also exhibit heterogeneity. It is necessary to expand the research on one cell line to other cell lines. At present, we think even adding another cell line will not fully confirm that it can be applied to more leukemia cells in general. Therefore, we will first report the interim results. In the future, we will screen multiple cell lines, such as myeloid or lymphoid cells, and combine clinical primary cell studies to confirm their universality. Of course, even if it is ultimately proven to have type specificity, it is still of great significance. So, would you be so kind to allow us to further implement your suggestions in the next works or article ?

Question 8: Introduction and Discussion are largely overlapping, I think that the authors should try not to be too repetitive. Even some experiments and how they are described in the Results are redundant (i.e. Fig. 3 A-B and Fig. 3F-G show the same results of Fig. 3C-D).

Answer: Based on your suggestions, we have made necessary modifications to the introduction and discussion sections to avoid overlapping as much as possible. Regarding the redundant and repetitive descriptions in the results, such as the same result descriptions in Figures 3A-B, 3F-G, and 3C-D, necessary deletions and modifications have also been made.

Question 9�Finally, in general, English language, typos and paragraph formatting should be checked.

Answer: Based on your suggestion, we have conducted a thorough language check and revision, as well as checked for spelling errors and paragraph formatting. Please review the new manuscript.

Question 10� Figure 2A and B, should be replaced with panels of better quality and higher magnification (the cell types are difficult to distinguish in the provided micrographs).

Answer: We have replaced Figures 2A and B with panels of better quality and higher magnification based on your suggestion The results regarding cell morphology are very stable in our induced differentiation system, and we have replaced them with higher resolution morphological maps about another repeated results. I am not sure if this meets the requirements. Please provide feedback and suggestions. Otherwise, we can provide it again.

6. PLOS authors have the option to publish the peer review history of their article (what does this mean?).

Question 1�If published, this will include your full peer review and any attached files.Answer: We choose “no”. However, our review may still be made public.

Thank you very much

Sincerely yours

Jiang

---

## [Decision Letter · Decision Letter 1]

18 Mar 2025

The up-regulation of TGF-β1 by miRNA-132-3p/WT1 is involved in inducing leukemia cells to differentiate into macrophages

PONE-D-24-22408R1

Dear Dr. Wang,

We’re pleased to inform you that your manuscript has been judged scientifically suitable for publication and will be formally accepted for publication once it meets all outstanding technical requirements.

Kind regards,

Francesco Bertolini, MD, PhD

Academic Editor

PLOS ONE

Additional Editor Comments (optional):

Reviewers' comments:

Reviewer's Responses to Questions

**Comments to the Author**

1. If the authors have adequately addressed your comments raised in a previous round of review and you feel that this manuscript is now acceptable for publication, you may indicate that here to bypass the “Comments to the Author” section, enter your conflict of interest statement in the “Confidential to Editor” section, and submit your "Accept" recommendation.

Reviewer #2: All comments have been addressed

Reviewer #3: All comments have been addressed

2. Is the manuscript technically sound, and do the data support the conclusions?

Reviewer #2: (No Response)

Reviewer #3: Yes

3. Has the statistical analysis been performed appropriately and rigorously?

Reviewer #2: (No Response)

Reviewer #3: Yes

4. Have the authors made all data underlying the findings in their manuscript fully available?

Reviewer #2: (No Response)

Reviewer #3: Yes

5. Is the manuscript presented in an intelligible fashion and written in standard English?

Reviewer #2: (No Response)

Reviewer #3: Yes

6. Review Comments to the Author

Reviewer #2: Plaese, check again the manuscript for typos and some mistakes. In particular, I think there is a mistake in the labelling of Figure 5C and D. Are the vectors indicated correctly?

Reviewer #3: In the manuscript “The up-regulation of TGF-β1 by miRNA-132-3p/WT1 is involved in inducing leukemia cells to differentiate into macrophages” Wang et al., study the effects of miRNA‑132‑3p, WT1, and TGF‑β1 in the phorbol ester – induced differentiation of THP-1 cells. They examine the mutual interference between the genes and their products, and conduct chromatin immunoprecipitation assays to determine the direct binding of WT1 on the promoter of TGF‑β1. The study is comprehensive and the English is understandable. Only minor proof reading may be needed.

7. PLOS authors have the option to publish the peer review history of their article (what does this mean? ). If published, this will include your full peer review and any attached files.

**Do you want your identity to be public for this peer review?** For information about this choice, including consent withdrawal, please see our Privacy Policy .

Reviewer #2: No

Reviewer #3: No

---

## [Editor Report · Acceptance letter]

PONE-D-24-22408R1

PLOS ONE

Dear Dr. Wang,

I'm pleased to inform you that your manuscript has been deemed suitable for publication in PLOS ONE. Congratulations! Your manuscript is now being handed over to our production team.

Kind regards,

on behalf of

Dr. Francesco Bertolini

Academic Editor

PLOS ONE